# Efficiency for Free: Ideal Data Are Transportable Representations

**Peng Sun**[1,2]    **Yi Jiang**[1]    **Tao Lin**[2,*]
[1]Zhejiang University    [2]Westlake University
sunpeng@westlake.edu.cn, yi_jiang@zju.edu.cn, lintao@westlake.edu.cn

## Abstract

Data, the seminal opportunity and challenge in modern machine learning, currently constrains the scalability of representation learning and impedes the pace of model evolution. In this work, we investigate the efficiency properties of data from both optimization and generalization perspectives. Our theoretical and empirical analysis reveals an unexpected finding: for a given task, utilizing a publicly available, task- and architecture-agnostic model (referred to as the 'prior model' in this paper) can effectively produce efficient data. Building on this insight, we propose the Representation Learning Accelerator (RELA), which promotes the formation and utilization of efficient data, thereby accelerating representation learning. Utilizing a ResNet-18 pre-trained on CIFAR-10 as a prior model to inform ResNet-50 training on ImageNet-1K reduces computational costs by $50\%$ while maintaining the same accuracy as the model trained with the original BYOL, which requires $100\%$ cost. Our code is available at: https://github.com/LINs-lab/ReLA.

## 1 Introduction

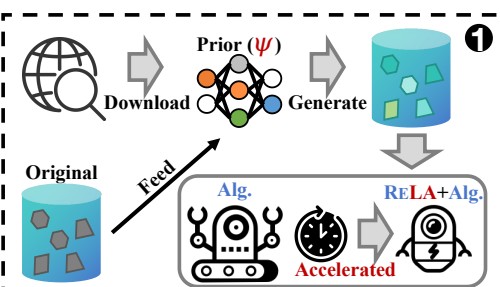 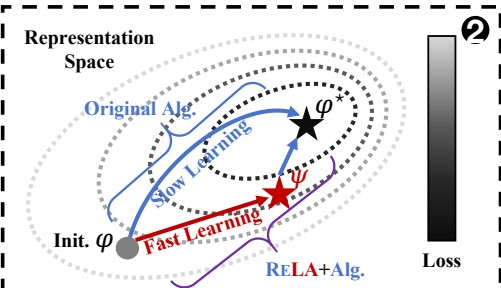

Figure 1: **Framework and Intuition of RELA**: (1) *Framework*: RELA serves as both a data optimizer and an auxiliary accelerator. Initially, it operates as a data optimizer by leveraging an dataset and a pre-trained model (e.g., one sourced from online repositories) to generate an efficient dataset. Subsequently, RELA functions as an auxiliary accelerator, enhancing existing (self-)supervised learning algorithms through the effective utilization of the efficient dataset, thereby promoting efficient representation learning. (2) *Intuition*: The central concept of RELA is to create an efficient-data-driven shortcut pathway within the learning process, enabling the initial model $\phi$ to rapidly converge towards a 'proximal representation $\psi$' of the target model $\phi^\star$ during the early stages of training. This approach significantly accelerates the overall learning process.

The available of massive datasets [20, 49] and recent advances in parallel data processing [28, 42] have facilitated the rapid evolution of large deep models, such as GPT-4 [1] and LVM [2]. These

---

*Corresponding author.

38th Conference on Neural Information Processing Systems (NeurIPS 2024).

models excel in numerous learning tasks, attributable to their impressive representation capabilities. However, the emergence of vast amounts of data within the modern deep learning paradigm raises two fundamental challenges: (i) *the demand for human annotations of huge datasets consumes significant social resources [45, 20, 43];* (ii) *training large models with increasing data and model capacity suffers from intensive computational burden [6, 16, 55].*

The community has made considerable efforts to enhance learning efficiency. Self-supervised learning methods [12, 69, 30, 10, 25, 14, 9, 3], with their superior representation learning devoid of human annotations via the self-learning paradigm, attempt to tackle the challenge (i). Concurrently, research has been conducted to mitigate data efficiency issues in challenge (ii): dataset distillation approaches [65, 56, 11, 72, 53, 54] have successfully synthesized a small distilled dataset, on which models trained on this compact dataset can akin to one trained on the full dataset.

However, challenges (i) and (ii) persist and yet are far from being solved [43, 45, 6, 16, 55], particularly the intervention of these two learning paradigms. In this paper, we identify two issues: (a) inefficiency in the self-supervised learning procedure compared to conventional supervised learning arises due to sub-optimal self-generating targets [30, 62]; (b) although training on the distilled dataset is efficient and effective, the distillation process of optimization-based approaches [11, 72, 37] is computationally demanding [18, 56], often surpassing the computational load of training on the full dataset. This limitation restricts its potential to accelerate representation learning. To tackle these challenges, we propose a novel open problem in the domain of representation learning:

> **Problem 1 (Accelerating Representation Learning through Free Models) .** *According to the No Free Lunch theorem [66], it is evident that accelerating the learning process without incorporating prior knowledge is inherently challenging. Fortunately, numerous publicly available pre-trained models can be accessed online, offering a form of free prior knowledge. Despite this, effectively utilizing these models poses several implicit challenges, as these models may not be directly relevant to our target learning task, or they may not be sufficiently well-trained. This leads to a question:*
>
> How can we leverage task- and architecture-agnostic publicly available models to accelerate representation learning for a specific task?

To address Problem 1 , we propose RELA to utilize a freely available model downloaded from the internet to generate efficient data for training. This approach aims to accelerate training during the initial stages by effectively leveraging these generated data, thereby establishing a rapid pathway for representation learning (see Figure 1 ). Specifically, we list our **five key contributions below** as the first step toward bridging representation learning with data-efficient learning:

(a) *Revealing beneficial/detrimental data properties for efficient/inefficient (self-)supervised learning (see Section 3.2 ).* We present a comprehensive analysis of linear models, demonstrating that data properties significantly influence the learning process by impacting the optimization of model training. Our findings reveal that modifications to the data can markedly enhance or impair this optimization. Additionally, we indicate that optimal training necessitates specific data properties—perfect bijective mappings between the samples and targets within a dataset.

(b) *Identifying the inefficiency problems of (self-)supervised learning from a data-centric perspective (see Section 3.3 ).* Specifically, we identify several factors contributing to the inefficiencies in (self-)supervised learning over real-world data. For instance, prevalent data augmentation techniques in modern deep learning can introduce a 'noisy mapping' issue, which may exacerbate the negative effects associated with inefficient data.

(c) *Generalization bound for models trained on optimized efficient data (see Section 3.4 ).* Although the efficiency properties of data do not inherently ensure the generalization of the trained model, i.e., efficient data alone cannot guarantee generalization ability, we present a generalization bound to analyze models trained on such data.

(d) *A novel method RELA to generate and exploit efficient data (see Section 4 ).* Leveraging our theoretical insights regarding the bounds of generalization and convergence rate, we introduce RELA, a novel optimization-free method tailored to efficiently generate and effectively exploit efficient data for accelerating representation learning.

(e) *An application of our RELA: accelerating (self-)supervised learning (see Section 5 and Appendix I ).* Extensive experiments across four widely-used datasets, seven neural network ar-

chitectures, eight self-supervised learning algorithms demonstrate the effectiveness and efficiency of RELA. Training models with RELA significantly outperforms training on the original dataset with the same budget, and even exceeds the performance of training on higher budget.

## 2  Related Work

This section integrates two distinct deep learning areas: (a) techniques to condense datasets while preserving efficacy; (b) self-supervised learning methods that enable training models on unlabeled data.

### 2.1  Dataset Distillation: Efficient yet Effective Learning Using Fewer Data

The objective of dataset distillation is to create a significantly smaller dataset that retains competitive performance relative to the original dataset.

**Refining proxy metrics between original and distilled datasets.** Traditional approaches involve replicating the behaviors of the original dataset within the distilled one. These methods aim to minimize discrepancies between surrogate neural network models trained on both synthetic and original datasets. Key metrics for this process include matching gradients [72, 32, 70, 44], features [63], distributions [71, 73], and training trajectories [11, 17, 22, 18, 68, 24]. However, these methods suffer from substantial computational overhead due to the incessant calculation of discrepancies between the distilled and original datasets. The optimization of the distilled dataset involves minimizing these discrepancies, necessitating multiple iterations until convergence. As a result, scaling to large datasets, such as ImageNet [20], becomes challenging.

**Extracting key information from original into distilled datasets.** A promising strategy involves identifying metrics that capture essential dataset information. These methods efficiently scale to large datasets like ImageNet-1K using robust backbones without necessitating multiple comparisons between original and distilled datasets. For instance, $SRe^2L$ [67] condenses the entire dataset into a model, such as pre-trained neural networks like ResNet-18 [26], and then extracts the knowledge from these models into images and targets, forming a distilled dataset. Recently, RDED [56] posits that images accurately recognized by strong observers, such as humans and pre-trained models, are more critical for learning.

**Summary.** We make the following observations regarding scalable dataset distillation methods utilizing various metrics: (a) a few of these metrics have proven effective for data distillation at the scale of ImageNet. (b) all these metrics require human-labeled data; (c) there is currently no established theory elucidating the conditions under which data distillation is feasible; (d) despite their success, the theory behind training neural networks with reduced data is underexplored.

### 2.2  Self-supervised Learning: Representation Learning Using Unlabeled Data

The primary objective of self-supervised learning is to extract robust representations without relying on human-labeled data. These representations should be competitive with those derived from supervised learning and deliver superior performance across multiple tasks.

**Contrasting self-generated positive and negative Samples.** Contrastive learning-based methods implicitly assign a one-hot label to each sample and its augmented versions to facilitate discrimination. Since InfoNCE [47], various works [25, 12, 13, 8, 31, 15, 74, 40, 9, 27] have advanced contrastive learning. MoCo [25, 13, 15] uses a momentum encoder for consistent negatives, effective for both CNNs and Vision Transformers. SimCLR [12] employs strong augmentations and a nonlinear projection head. Other methods integrate instance classification [8], data augmentation [31, 74], clustering [40, 9], and adversarial training [27]. These enhance alignment and uniformity of representations on the hypersphere [64].

**Asymmetric model-generating representations as targets.** Asymmetric network methods achieve self-supervised learning with only positive pairs [30, 50, 14], avoiding representational collapse through asymmetric architectures. BYOL [30] uses a predictor network and a momentum encoder. Richemond et al. [50] show BYOL performs well without batch statistics. SimSiam [14] halts the gradient to the target branch, mimicking the momentum encoder's effect. DINO [10] employs a self-distillation loss. UniGrad [59] integrates asymmetric networks with contrastive learning methods within a theoretically unified framework.

# 3 Revealing Critical Properties of Efficient Learning over Data

We begin by presenting formal definitions of supervised learning over a (efficient) dataset.

---

**Definition 1 (Supervised learning over data) .** *For a dataset $D = (D_X, D_Y) = \{(\mathbf{x}_i, \mathbf{y}_i)\}_{i=1}^{|D|}$, drawn from the data distribution $(X, Y)$ in space $(\mathcal{X}, \mathcal{Y})$, the goal of a model learning algorithm is to identify an optimal model $\phi^\star$ that minimizes the expected error defined by:*

$$\mathbb{E}_{(\mathbf{x}, \mathbf{y}) \sim (X, Y)} \left[ \ell(\phi^\star(\mathbf{x}), \mathbf{y}) \right] \leq \epsilon, \tag{1}$$

*where $\ell$ indicates the loss function and $\epsilon$ denotes a predetermined deviation. This is typically achieved through a parameterized model $\phi_{\boldsymbol{\theta}}$, where $\boldsymbol{\theta}$ denotes the model parameter within the parameter space $\boldsymbol{\Theta}$. The optimal parameter $\boldsymbol{\theta}^D$ is determined by training the model to minimize the empirical loss over the dataset:*

$$\boldsymbol{\theta}^D := \arg\min_{\boldsymbol{\theta} \in \boldsymbol{\Theta}} \{\mathcal{L}(\phi_{\boldsymbol{\theta}}; D; \ell)\} := \arg\min_{\boldsymbol{\theta} \in \boldsymbol{\Theta}} \left\{ \sum_{i=1}^{|D|} \ell(\phi_{\boldsymbol{\theta}}(\mathbf{x}_i), \mathbf{y}_i) \right\}. \tag{2}$$

*The training process leverages an optimization algorithm such as stochastic gradient descent [51, 33].*

---

**Definition 2 (Data-efficient Learning) .** *Data-efficient learning seeks to derive an optimized/efficient dataset, denoted as $S = (S_X, S_Y) = \{(\mathbf{x}_j, \mathbf{y}_j)\}_{j=1}^{|S|}$, from the original dataset $D$. The objective is to enable models $\phi_{\boldsymbol{\theta}^S}$ trained on $S$ to achieve the desired generalization performance, as defined in (1), with fewer training steps and a reduced computational budget compared to training on the original dataset $D$.*

---

## 3.1 Unifying (Self-)Supervised Learning from a Data-Centric Perspective

To ease the understanding and our methodology design in Section 3.4, we unify both conventional supervised learning and self-supervised learning as learning to map samples in $D_X$ to targets in $D_Y$: this view forms 'supervised learning' from a data-centric perspective. Specifically, these two learning paradigms involve generating targets $D_Y = \{\mathbf{y} \mid \mathbf{y} = \psi(\mathbf{x}) \text{ s.t. } \mathbf{x} \sim D_X\}$ and minimizing the empirical loss (2). The only difference lies in the target generation models (or simply labelers) $\psi$:

(a) *Conventional supervised learning*, referred to as human-supervised learning, generates targets via human annotation. Note that the targets are stored and used statically throughout the training.
(b) *Self-supervised learning* (also see Footnote 2), e.g., BYOL [30] utilizes an Exponential Moving Average (EMA) version of the learning model $\phi_{\boldsymbol{\theta}}$ to generate targets $\mathbf{y} = \text{EMA}[\phi_{\boldsymbol{\theta}}](\mathbf{x})$. Note that the targets are dynamically changing during training as the model $\phi_{\boldsymbol{\theta}}$ keeps evolving.

This unified perspective allows us to jointly examine and address the inefficient training issue of (self-)supervised learning from a data-centric perspective, in which in Section 3.2 we first study the impact of samples $D_X$ and targets $D_Y$ on the model training process and then investigate whether and how a distilled dataset $S = (S_X, S_Y)$ can facilitate this process.

## 3.2 Empirical and Theoretical Investigation of Data-Centric Efficient Learning

To elucidate the ideal data properties of training on a dataset $D$, we examine the simple task over a bimodal Gaussian mixture distribution as a case study. We begin by defining the problem.

---

**Definition 3 (Bimodal Gaussian mixture distribution) .** *Given two Gaussian distributions $\mathcal{N}_0(\mu_1, \Sigma^2 \mathbf{I})$ and $\mathcal{N}_1(\mu_2, \Sigma^2 \mathbf{I})$, where $\mu_1$ and $\mu_2$ are the means and $\Sigma^2$ is the variance (here we set $\mu_1 = 1$, $\mu_2 = 2$ and $\Sigma = 0.5$). We define a bimodal mixture data $G = (G_X, G_Y)$ as:*

$$G := \{(\mathbf{x}, y) \mid \mathbf{x} = (1-y) \cdot \mathbf{x}_0 + y \cdot \mathbf{x}_1\} \text{ s.t. } y \sim \text{Bernoulli}(p = 0.5), \mathbf{x}_0 \sim \mathcal{N}_0, \mathbf{x}_1 \sim \mathcal{N}_1. \tag{3}$$

*Moreover, we define a corresponding binary classification neural network model as:*

$$f_{\boldsymbol{\theta}}(\mathbf{x}) := \sigma \left( \boldsymbol{\theta}^{[1]} \cdot \text{ReLU} \left( \boldsymbol{\theta}^{[2]} \mathbf{x} + \boldsymbol{\theta}^{[3]} \right) + \boldsymbol{\theta}^{[4]} \right), \tag{4}$$

> *where $\sigma(z) = \frac{1}{1+e^{-z}}$ is the sigmoid activation function; $\text{ReLU}(z) = \max(0, z)$ is the activation function for the hidden layer, which provides non-linearity to the model; $\boldsymbol{\theta}^{[2]}$ and $\boldsymbol{\theta}^{[3]}$ are the weights and biases of the hidden layer; $\boldsymbol{\theta}^{[1]}$ and $\boldsymbol{\theta}^{[4]}$ are the weights and biases of the output layer.*

Modern representation learning fundamentally relies on optimization (see Definition 1 ). We show that modifications to the data can influence the convergence rate of the optimization process, thereby impacting the overall representation learning procedure. Furthermore, we try to uncover several key properties of data efficiency through our theoretical analysis of the case study. In the following, we denote the modified distribution by $G' = (G'_X, G'_Y)$ and examine the altered samples $G'_X$ and corresponding targets $G'_Y$ independently.

**Investigating the properties of modified samples.** The modification process here only rescales the variance of the original sample distribution $G_X$ defined in Definition 3 with new $\Sigma$ (rather than the default 0.5), while let $G'_Y := G_Y$; see explanations in Appendix E . Therefore, we examine the distilled samples $G'_X$ by setting the variable $\Sigma$ within the interval $(0, 1)$.

Results in Figure 2 demonstrate that the distilled samples $G'_X$ with smaller variance $\Sigma$ achieve faster convergence and better performance compared to that of $G$. To elucidate the underlying mechanism, we provide a rigorous theoretical analysis in Appendix B , culminating in Theorem 1 .

---

**Theorem 1 (Convergence rate of learning on efficient samples) .** *For the classification task stated in Definition 3 , the convergence rate for the model $f_{\boldsymbol{\theta}}$ trained $t$ after steps over distilled data $G'$ is:*
$$\mathbb{E}_{\boldsymbol{\theta}_t}\left[\mathcal{L}(f_{\boldsymbol{\theta}_t}; G'; \ell) - \mathcal{L}(f_{\boldsymbol{\theta}^\star}; G'; \ell)\right] \leq \tilde{\mathcal{O}}(\Sigma^2), \tag{5}$$
*where $\ell$ denotes the* MSE *loss, i.e., $\ell(\hat{y}, y) := \|\hat{y} - y\|^2$, and $f_{\boldsymbol{\theta}^\star}$ indicates the optimal model, $\tilde{\mathcal{O}}$ signifies the asymptotic complexity.* Modified samples characterized by a smaller value of $\Sigma$ facilitate faster convergence.

---

**Investigating the properties of modified targets.** On top of the property understanding for modified samples, we further investigate the potential of modified targets via $G'$. In detail, for modified samples, we consider the most challenging (c.f. Figure 2b ) yet the common case, namely $G'_X$ with $\Sigma = 1$ (see explanations in Appendix M ). For the corresponding modified targets $G'_Y$, similar to the prior data-efficient methods [56, 67], for any sample $\mathbf{x}$ drawn from $G'_X$, we refine its label by assigning $\hat{y} = \rho \cdot f_{\boldsymbol{\theta}^\star}(\mathbf{x}) + (1 - \rho) \cdot y$. Here, $\rho$ denotes the relabeling intensity coefficient, and $f_{\boldsymbol{\theta}^\star}$ represents a strong pre-trained model (simply, we utilize the model trained on the data in Figure 2c ).

---

**Theorem 2 (Convergence rate of learning on re-labeled data) .** *For the classification task as in Definition 3 , we have the convergence rate for the model $f_{\boldsymbol{\theta}}$ trained after $t$ steps over modified data $G'$:*
$$\mathbb{E}_{\boldsymbol{\theta}_t}\left[\mathcal{L}(f_{\boldsymbol{\theta}_t}; G'; \ell) - \mathcal{L}(f_{\boldsymbol{\theta}^\star}; G'; \ell)\right] \leq \tilde{\mathcal{O}}(1 - \rho). \tag{6}$$
*Note that $\rho$ controls the upper bound of the convergence rate, indicating that* using modified targets with a higher value of $\rho$ enables faster convergence.

---

Results in Figure 2 illustrate that the modified targets $G'_Y$ with higher values of $\rho$ lead to faster training convergence and better performance. See theoretical analysis in Theorem 2 and Appendix B .

### 3.3 Extended Understanding of Data-Centric Efficient Learning

The empirical and theoretical investigations regarding the properties of modified samples $G'_X$ and targets $G'_Y$ in Section 3.2 are limited to a simplified case (as described in Definition 3 ) and may not extend to all practical scenarios, such as training a ResNet [26] on the ImageNet dataset [20].

Interestingly, we observe that the advantageous modifications of both samples $G'_X$ and targets $G'_Y$ converge towards a unified principle: minimizing or preventing any sample $\mathbf{x}$ from being labeled with multiple or inaccurate targets $\mathbf{y}$. This principle emphasizes the importance of providing accurate and informative targets $\mathbf{y}$ for each sample $\mathbf{x}$, as analyzed in Remark 1 , and suggests extending this insight to any complex dataset like $S$.

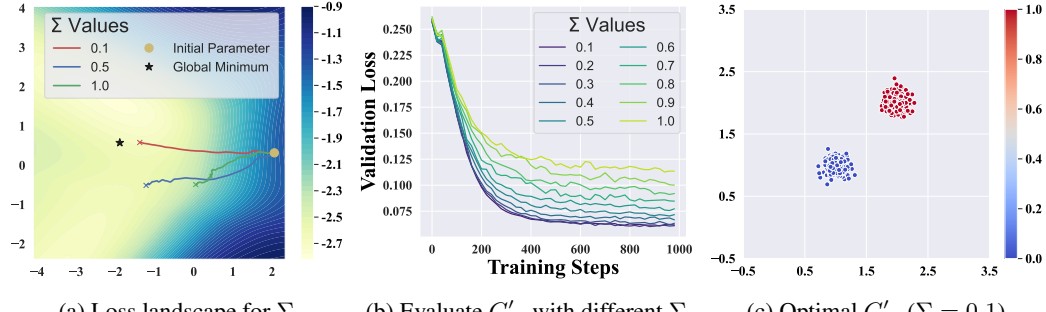

| (a) Loss landscape for $\Sigma$ | (b) Evaluate $G'_X$ with different $\Sigma$ | (c) Optimal $G'_X$ ($\Sigma = 0.1$) |

Figure 2: **Investigating modified samples with varied $\Sigma$ values**. Following [39], Figure 2a visualizes the validation loss landscape within a two-dimensional parameter space, along with three training trajectories corresponding to different $\Sigma$ settings. Figure 2b illustrates the performance of models trained using samples with varied $\Sigma$. The optimal case in our task, utilizing samples with $\Sigma = 0.1$ (which achieves the lowest validation loss in Figure 2b), is visualized in Figure 2c, where the color bar represents the values of targets $y$.

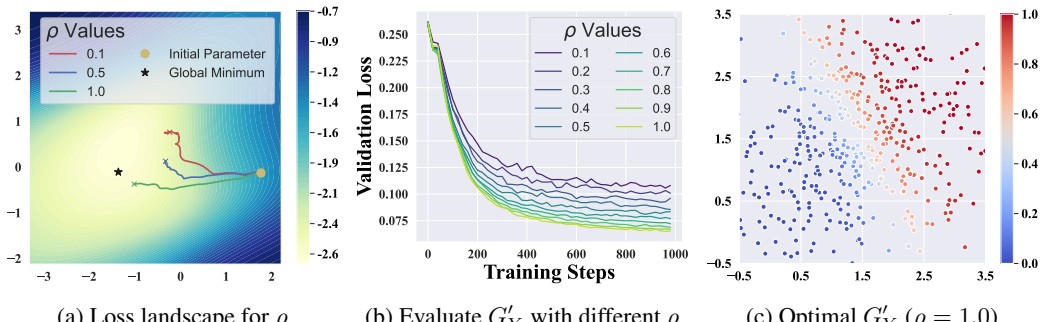

| (a) Loss landscape for $\rho$ | (b) Evaluate $G'_Y$ with different $\rho$ | (c) Optimal $G'_Y$ ($\rho = 1.0$) |

Figure 3: **Investigating modified targets with varied $\rho$ values**. We present a visualization of the validation loss landscape in Figure 3a, including three training trajectories that correspond to different $\rho$ settings. Figure 3b illustrates the performance of models trained using targets with varying $\rho$ values. The optimal scenario for our task, which uses targets with $\rho = 1.0$, is depicted in Figure 3c.

---

**Remark 1 (Ideal data properties avoid implicitly introduced gradient noise from data) .** *Intuitively, the semantic information within each sample* **x** *should be unique and not identical to another sample. Consequently, the exact target* **y***, which represents the semantic information of* **x***, should also be unique and informative. This implies the necessity of establishing bijective (or one-to-one) mappings between samples and their respective targets.*

*In contrast, when a sample* **x** *(or several similar samples) is labeled with multiple different targets* **y***, it may implicitly introduce noise into the gradient* $\nabla \ell(\mathbf{x}, \mathbf{y})$*, thereby hindering the optimization.*

---

However, real-world datasets often deviate from the ideal properties described above, as discussed in Remark 2 below and further analyzed in Appendix M .

---

**Remark 2 (Imperfect mappings and inaccurate targets in real-world datasets) .** *In practice, we observe that 'noisy mappings' between input samples and targets are prevalent in real-world datasets. As illustrated in Figure 6 , several common phenomena contribute to this issue:*

- *Similar or identical input samples may be assigned different targets due to using data augmentations, which is common in both self-supervised and human-supervised learning settings.*
- *Inaccurate targets may be generated, particularly in self-supervised learning scenarios.*
- *In human-supervised learning, all samples within a class are mapped to a one-hot target.*

*These imperfect mappings and inaccurate targets pose challenges to achieving optimal training efficiency and effectiveness for real-world datasets.*

---

### 3.4 Generalization-bounded Efficient Data Synthesis

Given insights from Remark 1 , an effective approach to generate efficient data $S$ from original data $D$ involves employing a high-quality labeler $\psi$ to relabel each sample **x** within $D$. This process

results in the formation of an optimized dataset $S$. However, the generalization ability of models trained on these optimized datasets $S$ is not inherently assured. For a given labeler $\psi$, we derive the generalization bound, which quantifies the representation distance between models $\phi_\theta$ and $\phi^\star$, trained on the optimized dataset $S$. This is presented in Theorem 3 (see Appendix D for the proof).

---

**Definition 4 (Representation distance[a]).** *We introduce our proposed metric as*

$$D_{\text{Rep}}(\phi_S \to \phi_T; D) := \inf_{\mathbf{W} \in \mathbb{R}^{m \times n}, \mathbf{b} \in \mathbb{R}^m} \{\mathbb{E}_{\mathbf{x} \sim D} [\ell(\mathbf{W}\phi_S(\mathbf{x}) + \mathbf{b}, \phi_T(\mathbf{x}))]\}, \tag{7}$$

*which quantifies the distance between a source model $\phi_S$ and a target model $\phi_T$ with respect to a dataset $D$, and the loss function is defined as $\ell(\hat{y}, y) := \mathbf{1}(\hat{y} \neq y)$.*

---

[a]Intuitively, a smaller $D_{\text{Rep}}(\phi_S \to \phi_T; D)$ indicates that the model $\phi_S$ can be transformed into $\phi_T$ over data $D$ via a linear model with relative ease. This also implies that $\phi_S$ can achieve the same linear evaluation performance as $\phi_T$ on data $D$.

---

**Theorem 3 (Generalization bound with labeler $\psi$).** *Assuming the model $\phi_\theta : \mathcal{X} \to \mathcal{Y}$ belongs to a hypothesis space $\Phi$. Then, for any $\delta \in (0, 1)$, with probability at least $1 - \delta$, we have*

$$D_{\text{Rep}}(\phi_\theta \to \phi^\star; X) \leq B_{\text{Sample}} + B_{\text{Target}} + B_{\text{Model}}, \tag{8}$$

*where* $B_{\text{Target}} = D_{\text{Rep}}(\psi \to \phi^\star; D_X)$, $B_{\text{Model}} = D_{\text{Rep}}(\phi_\theta \to \psi; S_X) + 2\mathfrak{R}_{D_X}(\Phi) + \tilde{\mathcal{O}}(|D_X|^{-1})$ *, and* $B_{\text{Sample}} = D_{\text{TV}}(S_X, D_X)$ *. $D_{\text{TV}}$ represents the total variation divergence [4], and $\mathfrak{R}_{D_X}$ is the empirical Rademacher complexity.*

---

Drawing insights from Theorem 1, 2 and 3, we define the properties of ideal data below:

---

**Definition 5 (Properties of ideal efficient data, including samples $S_X$ and targets $S_Y$).** *To meet the objectives of Definition 2, an ideal efficient data requires: ① Targets ($S_Y$):*

- *generating targets $\psi(\mathbf{x})$ from the labeler $\psi$ that are accurate (i.e., align with human-annotating targets[a] and optimal model $\phi^\star$), aiming to minimize $D_{\text{Rep}}(\psi \to \phi^\star; D_X)$ ;*
- *the generated target $\psi(\mathbf{x})$ should be informative to corresponding sample $\mathbf{x}$ (i.e., forming perfect bijective mappings with the original samples $\mathbf{x}$), according to Remark 5 ;*

*② Samples ($S_X$): low distribution disparity represented by $D_{\text{TV}}(S_X, D_X)$ and high sample diversity denoted as $|S_X|$, aiming at minimizing $D_{\text{TV}}(S_X, D_X)$ .*

---

[a]Alignment with human-annotated targets can occur via direct prediction or linear transportability.

---

To meet the requirement ② specified in Remark 5, we utilize *weak* data augmentation on $D_X$ to generate the set $S_X$, aimed at enhancing the diversity $|S_X|$ while ensuring the term $D_{\text{TV}}(S_X, D_X)$ remains minimal. However, satisfying requirement ① is non-trivial, as capturing a labeler $\psi$ that matches $\phi^\star$ is intractable (i.e., achieving $D_{\text{Rep}}(\psi \to \phi^\star; D_X) = 0$ is challenging).

Fortunately, our experiments in real-world scenarios, as discussed in Section A, demonstrate that employing a prior model as the labeler $\psi$ generally approximates $\phi^\star$ within the *representation space*. Therefore, we posit that introducing a prior model $\psi$ to generate an efficient dataset $S$ can typically accelerate the early stages of learning. Subsequently, the original dataset $D$ should be employed.

In (8), the last term, $D_{\text{Rep}}(\phi_\theta \to \psi; S_X) + 2\mathfrak{R}_{D_X}(\Phi) + \tilde{\mathcal{O}}(|D_X|^{-1})$ , includes $2\mathfrak{R}_{D_X}(\Phi)$ and $\tilde{\mathcal{O}}(|D_X|^{-1})$, which depend on the neural architecture and the size of $D_X$, respectively, and can be considered constants. Thus, optimizing the model involves minimizing $D_{\text{Rep}}(\phi_\theta \to \psi; S_X)$ through training $\phi_\theta$. A detailed technical solution is provided in Section 4.

## 4 Methodology

Building upon the theoretical insights from Section 3.2, we propose our RELA (see Figure 1): (a) RELA-D (🔧) is used to generate efficient data (c.f. Section 4.1); (b) RELA-F (⚡) guides the models to train over the efficient data from RELA-D (🔧) (c.f. Section 4.2).

## 4.1 RELA-D (🔧): Synthesis of Efficient Dataset

Motivated by two property requirements in Definition 5 , here we introduce our optimization-free synthesis process of both samples and targets in our RELA-D (see technical details in Appendix F ).

**Generating transportable representations as the targets.** We argue that *well-trained models (called* prior models*) on diverse real-world datasets using various neural network architectures and algorithms converge towards the same linear representation space. In other words, the generated pseudo representations $R_Y$ for samples $D_X$ using these prior models are linearly transportable to each other and to the human-annotating targets.* The empirical verifications refer to Appendix A . We further justify in Appendix F.1 that the requirement ① in Definition 5 can be achieved by employing a prior model as the ideal labeler $\psi : \mathbb{R}^d \to \mathbb{R}^m$, i.e., generating $R_Y = \{\psi(\mathbf{x}) \mid \mathbf{x} \sim D_X\}$ as the targets. The generation process of targets is conducted only once (refer to Appendix H for details), and the generated targets $R_Y$ are stored and combined with the samples $D_X$ to form the data $D = (D_X, R_Y)$.

**Efficient and distribution-aligned sample generation.** To satisfy requirement ② in Definition 5 efficiently, we employ basic data augmentations into data $D_X$ such as `RandomResizeCrop` with a minimum scale of 0.5 (as opposed to the default of 0.08) and `RandomHorizontalFlip` with $p = 0.5$.

## 4.2 RELA-F (⚡): Assist Learning with Generated Efficient Dataset

In this section, we showcase the significance of understanding ideal data properties and generated efficient dataset in assisting self-supervised learning, given this self-supervised paradigm on unlabeled data suffers from significant inefficiency issues compared to human-supervised learning [62].

Here we propose a plug-and-play method that can be seamlessly integrated into any existing self-supervised learning algorithm, significantly enhancing its training efficiency by introducing an additional loss term. Formally, the loss function is defined as follows:

$$\lambda \cdot \mathcal{L}_{\text{RELA}} + (1 - \lambda) \cdot \mathcal{L}_{\text{SSL}} , \text{ where } \mathcal{L}_{\text{RELA}} := \mathbb{E}_{\mathbf{x},\mathbf{y} \sim (D_X, R_Y)} \left[ \ell(\mathbf{W}\phi_{\boldsymbol{\theta}}(\mathbf{x}) - \mathbf{b}, \mathbf{y}) \right] , \quad (9)$$

where $\ell(\mathbf{z}, \mathbf{y}) := 1 - \mathbf{z} \cdot \mathbf{y}/(\|\mathbf{z}\|\|\mathbf{y}\|)$ be the loss function, $\mathcal{L}_{\text{SSL}}$ denotes the loss specified by any self-supervised learning method, respectively. Furthermore, the data $D = (D_X, R_Y)$ are collected using the strategy outlined in Section 4.1 , with updates occurring at each $k$-th epoch.

The dynamic coefficient $\lambda \in \{0, 1\}$ divides the training process into two distinct stages. Initially, $\lambda$ is set to 1 to emphasize $\mathcal{L}_{\text{RELA}}$, assuming its crucial role in the early learning phase. As the model $\phi_{\boldsymbol{\theta}}$ improves and self-generated targets become more reliable in $\mathcal{L}_{\text{SSL}}$, an adaptive attenuation algorithm adjusts $\lambda$ to 0 (note that the initial $\lambda$ is tuning-free for all cases and see Appendix J for details). As a result, only a single loss term in (9) is calculated, ensuring no extra computational cost with RELA.

To enhance the recognition of RELA-aided algorithms, we re-denote those that are used in their names. For example, the BYOL algorithm [30], when enhanced with RELA, is re-denoted as BYOL (⚡). Furthermore, as the prior models downloaded from the internet are not consistently robust, the aforementioned dynamic setting of $\lambda$ also prevents the model $\phi_{\boldsymbol{\theta}}$ from overfitting to potentially weak generated targets. The efficacy of our proposed RELA is empirically validated in Section 5 .

# 5 Experiments

This section describes the experimental setup and procedures undertaken to test our hypotheses and evaluate the effectiveness of our proposed methodologies.

**Experimental setting.** We list the settings below (see more details in Appendix K ).

● *Datasets:* For low-resolution data ($32 \times 32$), we evaluate our method on two datasets, i.e., CIFAR-10 [35] and CIFAR-100 [34]. For high-resolution data, we conduct experiments on two large-scale datasets including Tiny-ImageNet ($64 \times 64$) [36] and full ImageNet-1K ($224 \times 224$) [20], to assess the scalability and effectiveness of our method on more complex and varied datasets.

● *Neural network architectures:* Similar to prior works/benchmarks of dataset distillation [56] and self-supervised learning [57, 19], we use several backbone architectures to evaluate the generalizability

Table 1: **Benchmark our RELA with various prior models against BYOL**. We compare evaluation results of the models trained using ● BYOL with 10%, 20% and 50% training budget/steps; ● BYOL (⚡) with different prior models; ● BYOL with full budget, denoted as BYOL$^\star$ in this table. Regarding the prior models used for our RELA, we respectively utilize six models with increasing representation capabilities, including ● randomly initialized network (Rand.); ● four BYOL$^\star$-trained models (CF10-T, CF100-T, TIN-T, IN1K-T) corresponding to four datasets (listed below); ● CLIP-RN50. The evaluations are performed across four datasets, i.e., CIFAR-10 (CF-10), CIFAR-100 (CF-100), Tiny-ImageNet (T-IN), and ImageNet-1K (IN-1K). We underline the results that outperform the full training, and **bold** the results that achieve the highest performance using a specific ratio of budget. All the networks used for training are ResNet-18, except the ResNet-50 used for IN-1K.

| Dataset | % | BYOL | BYOL (⚡) w/ | | | | | | BYOL$^\star$ |
| --- | --- | --- | --- | --- | --- | --- | --- | --- | --- |
| | | | Rand. | CF10-T | CF100-T | TIN-T | IN1K-T | CLIP-RN50 | |
| CF-10 | 10 | $58.3 \pm 0.1$ | $71.4 \pm 0.0$ | $81.1 \pm 0.1$ | $78.2 \pm 0.1$ | $79.6 \pm 0.1$ | $81.6 \pm 0.0$ | $\mathbf{82.0 \pm 0.1}$ | $82.7 \pm 0.2$ |
| | 20 | $70.1 \pm 0.2$ | $77.1 \pm 0.2$ | $83.6 \pm 0.1$ | $81.4 \pm 0.0$ | $83.2 \pm 0.1$ | $\mathbf{84.4 \pm 0.1}$ | $83.9 \pm 0.1$ | |
| | 50 | $77.9 \pm 0.0$ | $82.7 \pm 0.1$ | $\underline{86.5 \pm 0.1}$ | $\underline{86.2 \pm 0.0}$ | $\underline{86.2 \pm 0.1}$ | $\mathbf{\underline{87.3 \pm 0.2}}$ | $\underline{86.7 \pm 0.0}$ | |
| CF-100 | 10 | $26.9 \pm 0.2$ | $41.8 \pm 0.2$ | $51.4 \pm 0.1$ | $51.4 \pm 0.1$ | $53.5 \pm 0.1$ | $\mathbf{56.4 \pm 0.2}$ | $55.4 \pm 0.1$ | $52.5 \pm 0.3$ |
| | 20 | $34.8 \pm 0.3$ | $48.1 \pm 0.1$ | $55.7 \pm 0.1$ | $55.7 \pm 0.1$ | $56.7 \pm 0.0$ | $\mathbf{\underline{59.5 \pm 0.1}}$ | $57.9 \pm 0.0$ | |
| | 50 | $41.4 \pm 0.3$ | $\underline{54.6 \pm 0.2}$ | $\underline{59.7 \pm 0.1}$ | $\underline{59.8 \pm 0.1}$ | $\underline{60.0 \pm 0.1}$ | $\mathbf{\underline{61.6 \pm 0.1}}$ | $\underline{61.0 \pm 0.0}$ | |
| T-IN | 10 | $25.1 \pm 0.3$ | $34.5 \pm 0.3$ | $39.0 \pm 0.1$ | $38.4 \pm 0.0$ | $41.2 \pm 0.1$ | $\mathbf{41.6 \pm 0.1}$ | $39.6 \pm 0.4$ | $43.6 \pm 0.3$ |
| | 20 | $30.7 \pm 0.1$ | $38.2 \pm 0.0$ | $41.9 \pm 0.0$ | $42.3 \pm 0.0$ | $43.2 \pm 0.1$ | $\mathbf{44.1 \pm 0.1}$ | $42.6 \pm 0.1$ | |
| | 50 | $37.7 \pm 0.2$ | $43.9 \pm 0.1$ | $45.6 \pm 0.1$ | $45.9 \pm 0.1$ | $45.8 \pm 0.1$ | $\mathbf{\underline{46.4 \pm 0.1}}$ | $46.3 \pm 0.1$ | |
| IN-1K | 10 | $44.5 \pm 0.1$ | $51.7 \pm 0.1$ | $53.7 \pm 0.1$ | $53.3 \pm 0.1$ | $53.6 \pm 0.1$ | $54.9 \pm 0.1$ | $\mathbf{56.2 \pm 0.1}$ | $61.9 \pm 0.1$ |
| | 20 | $55.3 \pm 0.0$ | $56.9 \pm 0.0$ | $57.6 \pm 0.1$ | $57.6 \pm 0.1$ | $57.8 \pm 0.1$ | $58.0 \pm 0.0$ | $\mathbf{59.5 \pm 0.1}$ | |
| | 50 | $60.8 \pm 0.2$ | $61.1 \pm 0.1$ | $\underline{62.1 \pm 0.1}$ | $61.8 \pm 0.1$ | $61.7 \pm 0.0$ | $61.9 \pm 0.0$ | $\mathbf{\underline{62.9 \pm 0.1}}$ | |

of our method, including ResNet-{18, 50, 101} [26], EfficientNet-B0 [58], MobileNet-V2 [52], ViT [21], and a series of CLIP-based models [49]. These architectures represent a range of model complexities and capacities, enabling a comprehensive assessment of our approach.

● *Baselines:* Referring to a prior widely-used benchmark [57, 19], we consider several state-of-the-art methods as baselines for a broader practical impact, including: SimCLR [12], Barlow Twins [69], BYOL [30], DINO [10], MoCo [25], SimSiam [14], SwAV [9], and Vicreg [3].

● *Evaluation:* Following previous benchmarks and research [57, 19, 12, 3], we evaluate all the trained models using offline linear probing strategy to reflect the representation ability of the trained models, and ensure a fair and comprehensive comparison with baseline approaches.

● *Implementation details:* We implement our method by extending a popular self-supervised learning open-source benchmark [57] and use their configurations therein. This includes using AdamW as the optimizer, with a mini-batch size of 128 (except for ImageNet-1K, where we use a mini-batch size of 512). We implement our method through PyTorch [48], and all experiments are conducted on NVIDIA RTX 4090 GPUs. See more detailed configurations and hyper-parameters in Appendix K .

## 5.1 Primary Experimental Results and Analysis

Recall that our RELA-D (🔧), as illustrated in Figure 1 and Section 4.1 , requires an unlabeled dataset and *any pre-trained model freely available online* to generate the efficient dataset. To justify the superior performance and generality of our RELA across various unlabeled datasets using prior models with different representation abilities, our comparisons in this subsection start with BYOL [30][2] and then extend to other self-supervised learning methods.

Table 1 demonstrates the efficacy and efficiency of our RELA in facilitating the learning of robust representations. Overall, *BYOL (⚡) consistently outperforms the original BYOL* when trained with a reduced budget. In certain cases, such as on CIFAR-100, BYOL (⚡) employing only 10% of the budget can surpass the performance of BYOL-trained models using the entire budget Specifically:

(a) A stronger prior model (e.g., CLIP) enhances the performance of RELA more effectively than a weaker model (e.g., Rand.);
(b) Our RELA is not sensitive to the prior knowledge. For instance, using CF10-T as the prior model can achieve competitive performance compared to that trained on extensive datasets (e.g., CLIP);

---

[2]Note that (1) BYOL is competitive across various datasets [30, 3, 57, 12], and (2) various self-supervised learning methods can be unified in the same framework [59] (see our detailed analysis in Appendix G ).

Table 2: **Evaluating our RELA on cross-architecture settings.** Our RELA-D (🔧) distills datasets with prior RN18 (Rand.) and CLIP-{RN101, RN50×4, ViT B/32, ViT B/16, ViT L/14}, then versus transfer to ResNet-18; MobileNet-V2; EfficientNet-B0; ViT T/16. We train models using 10% budget through (original) BYOL (⚡).

| Dataset | Arch. | Original | (⚡) w/ RN18 | RN101 | RN50x4 | ViT B/32 | ViT B/16 | ViT L/14 |
|---|---|---|---|---|---|---|---|---|
| CF-10 | ResNet-18 | $58.3 \pm 0.1$ | $71.4 \pm 0.0$ | $81.9 \pm 0.1$ | $82.1 \pm 0.3$ | $83.2 \pm 0.2$ | $83.1 \pm 0.1$ | $82.4 \pm 0.1$ |
| | MobileNet-V2 | $47.7 \pm 0.1$ | $69.4 \pm 0.0$ | $82.2 \pm 0.1$ | $80.8 \pm 0.0$ | $81.6 \pm 0.1$ | $82.9 \pm 0.2$ | $81.2 \pm 0.2$ |
| | EfficientNet-B0 | $23.9 \pm 0.2$ | $68.8 \pm 0.6$ | $83.2 \pm 0.2$ | $83.9 \pm 0.1$ | $87.4 \pm 0.1$ | $86.4 \pm 0.1$ | $83.1 \pm 0.1$ |
| | ViT T/16 | $43.4 \pm 0.1$ | $57.1 \pm 0.1$ | $65.9 \pm 0.0$ | $66.4 \pm 0.1$ | $69.9 \pm 0.3$ | $68.8 \pm 0.1$ | $63.7 \pm 0.1$ |
| T-IN | ResNet-18 | $25.1 \pm 0.3$ | $34.5 \pm 0.3$ | $38.3 \pm 0.1$ | $39.1 \pm 0.4$ | $35.8 \pm 0.1$ | $32.4 \pm 0.1$ | $28.4 \pm 0.2$ |
| | MobileNet-V2 | $8.8 \pm 0.1$ | $28.3 \pm 0.3$ | $39.9 \pm 0.1$ | $36.8 \pm 0.2$ | $36.0 \pm 0.0$ | $37.9 \pm 0.3$ | $20.6 \pm 0.5$ |
| | EfficientNet-B0 | $4.1 \pm 0.0$ | $33.2 \pm 0.3$ | $43.5 \pm 0.2$ | $41.7 \pm 0.2$ | $44.0 \pm 0.1$ | $44.2 \pm 0.0$ | $37.9 \pm 0.1$ |
| | ViT T/16 | $12.5 \pm 0.1$ | $24.6 \pm 0.0$ | $26.1 \pm 0.1$ | $27.6 \pm 0.1$ | $26.9 \pm 0.2$ | $24.5 \pm 0.1$ | $21.6 \pm 0.0$ |

Table 3: **Evaluating our RELA across different self-supervised learning methods.** We extend our analysis beyond BYOL by training and evaluating models using seven additional self-supervised learning methods, along with their RELA-augmented counterparts (⚡), utilizing randomly initialized ResNet-18 (Rand.) and CLIP-RN50 as prior models for the RELA-D (🔧). All methods are trained using 10% budget.

| Dataset | Method | SimCLR | Barlow | DINO | MoCo | SimSiam | SwAV | Vicreg |
|---|---|---|---|---|---|---|---|---|
| CF-10 | Original | $70.7 \pm 0.2$ | $63.7 \pm 0.3$ | $66.2 \pm 0.2$ | $67.4 \pm 0.4$ | $45.8 \pm 0.4$ | $66.2 \pm 0.3$ | $71.3 \pm 0.2$ |
| | (⚡) w/ Rand. | $70.9 \pm 0.0$ | $68.8 \pm 0.2$ | $70.6 \pm 0.1$ | $70.9 \pm 0.1$ | $66.7 \pm 0.1$ | $69.5 \pm 0.2$ | $71.3 \pm 0.1$ |
| | CLIP-RN50 | $76.4 \pm 0.1$ | $76.5 \pm 0.2$ | $82.4 \pm 0.1$ | $79.8 \pm 0.1$ | $79.3 \pm 0.1$ | $77.3 \pm 0.0$ | $80.1 \pm 0.1$ |
| T-IN | Original | $30.4 \pm 0.1$ | $28.9 \pm 0.4$ | $26.7 \pm 0.3$ | $27.1 \pm 0.2$ | $17.8 \pm 0.3$ | $20.2 \pm 0.1$ | $34.0 \pm 0.1$ |
| | (⚡) w/ Rand. | $30.7 \pm 0.2$ | $31.9 \pm 0.1$ | $29.4 \pm 0.2$ | $33.4 \pm 0.1$ | $25.4 \pm 0.1$ | $29.1 \pm 0.2$ | $34.1 \pm 0.1$ |
| | CLIP-RN50 | $33.0 \pm 0.3$ | $33.5 \pm 0.2$ | $35.1 \pm 0.0$ | $37.1 \pm 0.1$ | $32.6 \pm 0.1$ | $32.6 \pm 0.1$ | $39.1 \pm 0.2$ |

(c) A randomly initialized model can effectively aid in accelerating learning through our RELA. This can be considered an effective scenario of "weak-to-strong supervision" [7] using pseudo targets.

**Cross-architecture generalization.** RELA-D (🔧) generates efficient datasets using a specific neural architecture. To evaluate the generalization ability of these datasets, it is essential to test their performance on various architectures not used in the distillation process. Table 2 presents the performance of our RELA in conjunction with various prior models and trained model architectures, demonstrating its robust generalization ability. Specifically:

(a) The integration of RELA always enhances the performance of original BYOL;
(b) Our RELA method exhibits minimal sensitivity to the architecture of the prior model, as evidenced by the comparable performance of BYOL (⚡) using both ViT-based and ResNet-based models.

**Combining RELA across various self-supervised learning methods.** To demonstrate the effectiveness and versatility of RELA in enhancing various self-supervised learning methods, we conduct experiments with widely-used techniques. Table 3 presents the results, highlighting the robust generalization capability of RELA. Our findings consistently show that RELA improves the performance of these methods while maintaining the same budget ratio, emphasizing its potential on learning using unlabeled data. Additionally, we provide the results when combining RELA with human-supervised learning in Appendix I .

## 6   Conclusion and Limitation

In this paper, to address the Problem 1 , we investigate the optimal properties of data, including samples and targets, to identify the properties that improve generalization and optimization in deep learning models. Our theoretical insights indicate that targets which are informative and linearly transportable to strong representations (e.g., human annotations) enable trained models to exhibit robust representation abilities. Furthermore, we empirically find that well-trained models (called prior models) across various tasks and architectures serve as effective labelers for generating such targets. Consequently, we propose the Representation Learning Accelerator (RELA), which leverages any freely available prior model to generate high-quality targets for samples. Additionally, RELA can enhance existing (self-)supervised learning approaches by utilizing these generated data to accelerate training. However, our theoretical analysis is restricted to the simplified scenario described in Definition 3 , which has limited applicability in real-world contexts.

# 7 Acknowledgement

We thank Xinyi Shang, Bowen Ding and and the anonymous reviewers for their invaluable comments and feedback. We also thank Bei Shi for assisting with the partial code implementation. This work was supported in part by the National Science and Technology Major Project (No. 2022ZD0115101), the Research Center for Industries of the Future (RCIF) at Westlake University, and the Westlake Education Foundation.

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

# Contents

# A   Ablation Study

We conduct ablation studies to understand the impact of each component of RELA on performance.

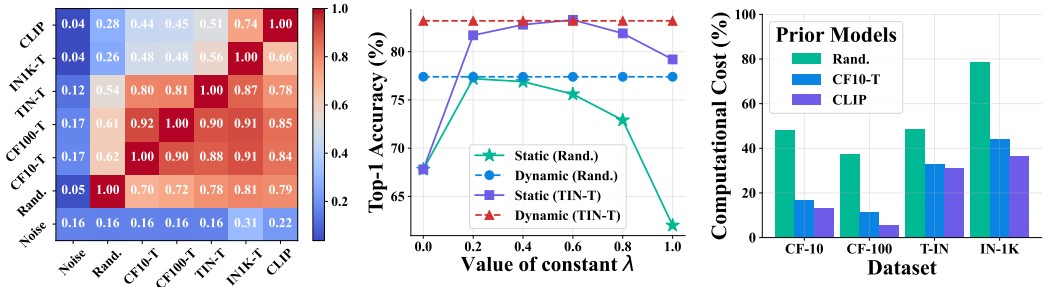

(a) Representation similarity      (b) Dynamic vs. static strategies      (c) Computational cost analysis

Figure 4: **Ablation study on BYOL (⚡) components and parameters**. (a) We analyze the representation similarity between various source models (indicated on the x-axis) and target models (indicated on the y-axis). (b) We compare the static RELA weight setting strategy with our adaptive strategy. Dotted lines ('- -') represent our adaptive strategy, while solid lines ('—') denote the static $\lambda$ setting strategy. Specifically, in the static weight setting (e.g., 0.4), the first 40% of the training leverages RELA, with the remaining 60% employing the original algorithm. (c) We present the computational cost, quantified as training time/steps, of our RELA across various prior models.

**Empirical representation similarity.** Our foundational assumption of our RELA is that the representations of well-trained models, developed using various neural network architectures and algorithms on diverse real-world datasets, exhibit linear transportability to one another. To test this hypothesis, we assess representation similarity, a metric that quantifies the linear transferability between pre-trained models. The results, depicted in Figure 4a, demonstrate that representations from robust models (e.g., CLIP) can be effectively transferred to less robust models (e.g., CF10-T). This finding aligns with our results in Table 1, showing that leveraging powerful models (e.g., CLIP and IN1K-T) consistently enhances learning in models trained on datasets with limited knowledge, such as CF-10.

**Combining RELA and BYOL with static $\lambda$ setting strategies.** The coefficient $\lambda$, as introduced in Section 4.2, is pivotal in controlling the weight of the RELA phase during training. To assess the robustness of our adaptive strategy, which dynamically adjusts $\lambda$, we compare it to a static $\lambda$ setting strategy. The results in Figure 4b indicate that larger (smaller) RELA weights are advantageous when using a strong (weak) prior model. Nonetheless, static settings lack generalizability across various scenarios, whereas our adaptive strategy demonstrates superior generalization capabilities.

**Analysis of the computational cost when using RELA with different prior models.** To validate that our RELA, in conjunction with various prior models, can effectively reduce computational costs in self-supervised learning, we conducted experiments comparing the computational expense required for BYOL (⚡) to achieve equivalent performance to the original BYOL trained with a full budget. The results, illustrated in Figure 4c, consistently demonstrate that RELA assists BYOL in lowering training costs while maintaining equivalent performance levels. Furthermore, it is evident that employing robust prior models consistently leads to greater reductions in training budgets.

# B    Proof of Theorem 1

In this section, we prove a slightly modified version of Theorem1, extending the distribution to Generalized Gaussian Mixture(GGM) and making some assumptions for technical reasons. Yet this proof could still reflect the essential of the theorem.

## B.1    Setup

**Notation** $N(\mu, \alpha, \beta)$ denotes the generalized Gaussian distribution with pdf $\frac{\beta}{2\alpha\Gamma(1/\beta)}e^{-(|x-\mu|/\alpha)^\beta}$, B for Bernoulli distribution.
We focus on the 1-dim situation. Assume that $\mu_1 < \mu_2$. Define the original data distribution($\mathcal{N}_0 = N(\mu_1, \alpha_0, \beta_0)$ and $\mathcal{N}_1 = N(\mu_2, \alpha_0, \beta_0)$)

$$G := \{(x, y) \mid y \sim 2 \cdot B(1, \frac{1}{2}) - 1, x \sim \frac{1-y}{2} \cdot \mathcal{N}_0 + \frac{1+y}{2} \cdot \mathcal{N}_1\}$$

and the modified one ($\mathcal{N}_0' = N(\mu_1, \alpha, \beta)$ and $\mathcal{N}_1' = N(\mu_2, \alpha, \beta)$):

$$G' := \{(x, y) \mid y \sim 2 \cdot B(1, \frac{1}{2}) - 1, x \sim \frac{1-y}{2} \cdot \mathcal{N}_0' + \frac{1+y}{2} \cdot \mathcal{N}_1'\}.$$

Our task is predicting $y$ given $x$. Note that $y \in \{\pm 1\}$, which is a bit different from the definition in Section 3.2. In 1-dim situation, we just need one parameter for this classification task, so define $f_\theta(x) := \text{sign}(x + \theta)$ to fit the distribution. We could compute the generalization loss on original distribution:

$$\mathcal{L}(f_\theta) = (\int_{-\theta}^{+\infty} dF_- + \int_{-\infty}^{-\theta} dF_+)/2 = (1 - \int_{-\frac{\theta+\mu_2}{\alpha_0}}^{-\frac{\theta+\mu_1}{\alpha_0}} dF)/2$$

Obviously $\theta^\star = -\frac{\mu_1+\mu_2}{2}$, we have:

$$\mathcal{L}(f_\theta) - \mathcal{L}(f_{\theta^\star}) = (\int_{-\frac{\mu_2-\mu_1}{2\alpha_0}}^{\frac{\mu_2-\mu_1}{2\alpha_0}} dF - \int_{-\frac{\theta+\mu_2}{\alpha_0}}^{-\frac{\theta+\mu_1}{\alpha_0}} dF)/2$$
$$\leq C_1 \cdot (\theta - \theta^\star)^2 \quad (or\ C_1'\ |\theta - \theta^\star|)$$

where $C_1$, $C_1'$ are constants, $F_0$ $F_1$ $F$ denote the CDF of $\mathcal{N}_0$ $\mathcal{N}_1$ and $N(0, 1, \beta_0)$ respectively. The inequality above is due to the fact that function $h(x) = (\int_{-1}^1 dF - \int_{x-1}^{x+1} dF)/x^2$ has limits at 0 and so is bounded.

## B.2    Algorithm

For a dataset $\{(x_i, y_i)\}_{i=1}^n$, set the loss function $L(\theta) = \frac{1}{n} \sum_{i=1}^n \ell [y_i(x_i + \theta)]$, $\ell(v) = \frac{1}{2}(1 - v)^2$. We apply the stocastic gradient descent algorithm and assume the online setting ($n = 1$): at step $t$ draw one sample $(x_t, y_t)$ from $G'$ then use the gradient $\nabla L(\theta_t)$ to update $\theta$ ($\eta \in (0, 1), t \in \mathbb{N}$):

$$\theta_{t+1} = \theta_t - \eta \nabla L(\theta_t),$$
$$\nabla L(\theta_t) = \theta + (x_t - y_t).$$

It can be observed that randomness of $x$ leads to noies on gradient.

## B.3    Bounds with Variance

We prove the proposition that lower variance of GG can make convergence faster, i.e. $\mathbb{E}[\mathcal{L}(f_{\theta_t}) - \mathcal{L}(f_{\theta^\star})]$ is bounded by an increasing function of variance ($t$ fixed).

*Proof.* From above, we could get

$$\theta_t = (1 - \eta)^t \theta_0 - \eta \left[(x_{t-1} - y_{t-1}) + (1 - \eta)(x_{t-2} - y_{t-2}) + \cdots + (1 - \eta)^{t-1}(x_0 - y_0)\right]$$

and so :

$$\mathbb{E}\left[\mathcal{L}(f_{\theta_t}) - \mathcal{L}(f_{\theta^\star})\right] \le C_1 \mathbb{E}\left[(\theta_t - \theta^\star)^2\right]$$

$$= C_1 \mathbb{E}\left\{\left[(1-\eta)^t(\theta_0 - \theta^\star) - \eta \sum_{j=1}^{t}(1-\eta)^{j-1}(x_{t-j} - y_{t-j} + \theta^\star)\right]^2\right\}$$

$$= C_1 \mathbb{E}\left[(1-\eta)^{2t}(\theta_0 - \theta^\star)^2 + \eta^2 \sum_{j=1}^{t}(1-\eta)^{2(j-1)}(x_{t-j} - y_{t-j} + \theta^\star)^2\right]$$

$$= C_1 \left((1-\eta)^{2t}(\theta_0 - \theta^\star)^2 + \frac{\eta}{(2-\eta)}(1-(1-\eta)^{2t})\left[\frac{\alpha^2 \Gamma(3/\beta)}{\Gamma(1/\beta)} + \left(1 - \frac{\mu_2 - \mu_1}{2}\right)^2\right]\right)$$

The last two equalities is due to the fact that for $(x, y) \sim G'$

$$\mathbb{E}\left[x - y + \theta^\star\right] = 0 \ ,$$

$$\mathbb{E}\left[(x - y + \theta^\star)^2\right] = \frac{\alpha^2 \Gamma(3/\beta)}{\Gamma(1/\beta)} + \left(1 - \frac{\mu_2 - \mu_1}{2}\right)^2 .$$

$\square$

## B.4 Nonlinear case

In this subsection, we conduct some qualitative analysis on the nonlinear case. The setting is the same as that in Section 3.2. We point out the differences compared with the linear case above: $\mathbf{x} \in \mathbb{R}^d, y \in \{0, 1\}$ and

$$f_{\boldsymbol{\theta}}(\mathbf{x}) := \sigma\left(\boldsymbol{\theta}^{[1]} \cdot \mathrm{ReLU}\left(\boldsymbol{\theta}^{[2]}\mathbf{x} + \boldsymbol{\theta}^{[3]}\right) + \boldsymbol{\theta}^{[4]}\right)$$

where $\sigma(z) = \frac{1}{1+e^{-z}}$ is the sigmoid function; $\mathrm{ReLU}(z) = \max(0, z)$ is the activation function for the hidden layer, which provides non-linearity to the model; $\boldsymbol{\theta}^{[2]}$ and $\boldsymbol{\theta}^{[3]}$ are the weights and biases of the hidden layer; $\boldsymbol{\theta}^{[1]}$ and $\boldsymbol{\theta}^{[4]}$ are the weights and biases of the output layer.

To make things explicit, we still assume the online setting and set the loss function $L(\theta) = \frac{1}{2}(f_{\boldsymbol{\theta}}(\mathbf{x}) - y)^2$. Assume after some iterations, $\boldsymbol{\theta}^{[2]} \cdot \mu_1 + \boldsymbol{\theta}^{[3]} < 0$ and $\boldsymbol{\theta}^{[2]} \cdot \mu_2 + \boldsymbol{\theta}^{[3]} > 0$ (coordinate-wise). In this situation, we could see that if $\mathbf{x}$ is close to its mean($\mu_1$ or $\mu_2$), the sign of $\mathrm{ReLU}\left(\boldsymbol{\theta}^{[2]}\mathbf{x} + \boldsymbol{\theta}^{[3]}\right)$ will be the same as $y$. So $f_{\boldsymbol{\theta}}$ will become an optimal classifier if $\boldsymbol{\theta}^{[1]} \to +\infty$ and $\boldsymbol{\theta}^{[4]} \to -\infty$. We focus on $\boldsymbol{\theta}^{[1]}$, using SGD:

$$\boldsymbol{\theta}_{t+1}^{[1]} = \boldsymbol{\theta}_t^{[1]} - \eta \frac{\partial L}{\partial \boldsymbol{\theta}^{[1]}},$$

$$\frac{\partial L}{\partial \boldsymbol{\theta}^{[1]}} = (f_{\boldsymbol{\theta}}(\mathbf{x}) - y)\sigma(1 - \sigma)\mathrm{ReLU}\left(\boldsymbol{\theta}^{[2]}\mathbf{x} + \boldsymbol{\theta}^{[3]}\right)$$

Note that we drop the variable value in $\sigma(\cdot)$ to make the expression more compact.

Then we can analyze the phenomenon qualitatively: larger $\Sigma$ will make convergence slower. The reason is that the larger $\Sigma$ is, when $\mathbf{x}$ is drawn from $\mathcal{N}_1$ ($y = 1$), $\boldsymbol{\theta}^{[2]}\mathbf{x} + \boldsymbol{\theta}^{[3]} < 0$ is more likely to happen(i.e. straying far away from the mean), causing $\boldsymbol{\theta}^{[1]}$ to stop updating; what's worse, when $\mathbf{x}$ is drawn from $\mathcal{N}_0$ ($y = 0$), with larger probability $\boldsymbol{\theta}^{[2]}\mathbf{x} + \boldsymbol{\theta}^{[3]} > 0$ which will make $\boldsymbol{\theta}^{[1]}$ to go in the opposite direction. In summary, it is $\Sigma$ that makes the gradient noisy thus impacts the convergence rate.

## B.5 From the Perspective of Feature Learning

In essence, the theoretical results in [41] could also be interpreted as a proof of the theorem. [41] study the learning of a two-layer ReLU neural network for $k$-class classification via stochastic gradient descent (SGD), assuming that each class corresponds to $l$ patterns(distributions), with every two of the $k \times l$ distributions of the input data are separated by a distance $\delta$. Below is the main theorem in [41]:

**Proposition 1** . *Suppose some assumptions are satisfied, then for every $\epsilon > 0$, there is $M = poly(k, l, 1/\delta, 1/\epsilon)$ such that for every $m \geq M$, after doing a minibatch SGD with batch size $B = poly(k, l, 1/\delta, 1/\epsilon, \log m)$ and learning rate $\eta = \frac{1}{m \cdot poly(k, l, 1/\delta, 1/\epsilon, \log m)}$ for $T = poly(k, l, 1/\delta, 1/\epsilon, \log m)$ iterations, with high probability, the generalization error(the probability that the model misclassifies) of the learned model is at most $\epsilon$.*

Theoretical results above show that a larger $\delta$ helps network to learn more efficiently. (In our case, $\delta$ can be roughly viewed as the Mahalanobis distance between the two Gaussians, which is inverse proportion to the variance.) Also, Appendix D.2 of [41] demonstrates an example very similar to ours in which the input data is drawn from Gaussian distributions with different means, indicating increasing variance of the Gaussian causes the test accuracy to decrease and takes longer time to get a good solution.

## C    Proof of Theorem 2

### C.1    Setting

Use the same setting in Section B(linear case) except that

$$G' := \left\{ (x, y') \mid y \sim 2 \cdot \mathrm{B}(1, \tfrac{1}{2}) - 1, x = \frac{1-y}{2} \cdot \mathcal{N}_0 + \frac{1+y}{2} \cdot \mathcal{N}_1, y' = \rho \cdot f_{\theta^*}(x) + (1-\rho)y \right\}.$$

In other words, we modify the distribution of $y$ instead of $x$ this time.

### C.2    Bounds with $\rho$

We're going to prove that higher $\rho$ can make convergence faster, i.e. $\mathbb{E}\left[\mathcal{L}(f_{\theta_t}) - \mathcal{L}(f_{\theta^*})\right]$ is bounded by an decreasing function of $\rho$ ($t$ fixed).

*Proof.* The crucial part $x - y' + \theta^\star = \rho(x - f_{\theta^*}(x) + \theta^\star) + (1-\rho)(x - y + \theta^\star)$, and in fact

$$\mathbb{E}|x - y + \theta^\star| - \mathbb{E}|x - f_{\theta^*}(x) + \theta^\star| := \epsilon_0 > 0 \ .$$

Similarly, we can get bounds with $\rho$ (see $C_1'$ in Section B):

$$
\begin{aligned}
\mathbb{E}\left[\mathcal{L}(f_{\theta_t}) - \mathcal{L}(f_{\theta^*})\right] &\leq C_1'\mathbb{E}\left[(1-\eta)^t|\theta_0 - \theta^\star| + (1 - (1-\eta)^t)|x - y' + \theta^\star|\right] \\
&\leq C_2(1-\eta)^t + C_1'(1 - (1-\eta)^t)\left[\rho \cdot \mathbb{E}|x - f_{\theta^*}(x) + \theta^\star| \right. \\
&\quad \left. + (1-\rho) \cdot \mathbb{E}|x - y + \theta^\star|\right] \\
&\leq C_2(1-\eta)^t + (1 - (1-\eta)^t)(C_3 - C_4\rho)
\end{aligned}
$$

where $C_2 = C_1' \, |\theta_0 - \theta^\star|$, $C_3 = C_1' \, \mathbb{E}|x - y + \theta^\star|$, $C_4 = C_1' \, \epsilon_0 > 0$. $\qquad\square$

### C.3    Nonlinear case

To see the impact of modifying $y$ more clearly, we directly set $\rho = 1$ and conduct a similar analysis as in Section B.4. Let's still focus on $\theta^{[1]}$ and use the same assumptions in Section B.4, then we have:

$$\theta^{[1]}_{t+1} = \theta^{[1]}_t - \eta\frac{\partial L}{\partial \theta^{[1]}},$$

$$\frac{\partial L}{\partial \theta^{[1]}} = (f_{\theta}(\mathbf{x}) - f_{\theta^*}(\mathbf{x}))\sigma(1 - \sigma)\mathrm{ReLU}\left(\theta^{[2]}\mathbf{x} + \theta^{[3]}\right)$$

Note $y$ is replaced by $f_{\theta^*}(\mathbf{x})$. For instance $\mathbf{x}$ is drawn from $\mathcal{N}_0$ ($y = 0$) but strays far away from the $\mu_1$, causing $\theta^{[2]}\mathbf{x} + \theta^{[3]} > 0$. In this situation $f_{\theta^*}$ is likely to regard $\mathbf{x}$ as a sample from $\mathcal{N}_1$ (i.e. $f_{\theta^*}(\mathbf{x})$ close to 1) thus making $\theta^{[1]}$ to go in the right direction instead of the opposite. This explains why larger $\rho$ can make convergence faster.

# D  Proof of Theorem 3

We follow some proof steps in [5]. Let's begin by introducing some notations used in this section.
**Notation and Setup** $\mathcal{X}$ is the input space, $\mathcal{D}_S$ and $\mathcal{D}_T$ are two distributions over $\mathcal{X}$. Let $\mathcal{Y} = \{0, 1\}$. $\mathcal{H}$ denotes a hypothesis class from $\mathcal{X}$ to $\mathcal{Y}$. To simplify notations, $\forall h, f \in \mathcal{H}$ let $\epsilon_S(h, f) = E_{\mathbf{x} \sim \mathcal{D}_S}[\mathbf{1}(h(\mathbf{x}) \neq f(\mathbf{x}))]$, and $\hat{\epsilon}_S(h, f)$ be empirical error ($\epsilon_T(h, f)$, $\hat{\epsilon}_T(h, f)$ similar).

Then we introduce some concepts and lemmas, most of which are from [5].

**Definition 6 ($\mathcal{H}$-divergence) .** *The $\mathcal{H}$-divergence between two distributions $\mathcal{D}$ and $\mathcal{D}'$ is defined as:*
$$d_{\mathcal{H}}(\mathcal{D}, \mathcal{D}') = 2 \sup_{h \in \mathcal{H}} |\Pr_{\mathcal{D}}[I(h)] - \Pr_{\mathcal{D}'}[I(h)]|$$
*where $I(h) = \{\mathbf{x} : h(\mathbf{x}) = 1\}$.*

**Definition 7 (Total Variation Distance) .** *For two distributions $\mathcal{D}$ and $\mathcal{D}'$, the total variation distance of them is defined as:*
$$D_{\mathrm{TV}}(\mathcal{D}, \mathcal{D}') = \sup_{A \subseteq \mathcal{F}} |\Pr_{\mathcal{D}}(A) - \Pr_{\mathcal{D}'}(A)|$$
*where $\mathcal{F}$ denotes the collection of all events in the probability space.*

**Lemma 1 .** *For two distributions $\mathcal{D}$ and $\mathcal{D}'$, by definition it's easy to see that:*
$$\frac{1}{2} d_{\mathcal{H}}(\mathcal{D}, \mathcal{D}') \leq D_{\mathrm{TV}}(\mathcal{D}, \mathcal{D}')$$

**Definition 8 (Symmetric Difference Hypothesis Space) .**
$$\mathcal{H} \Delta \mathcal{H} := \{g : \mathcal{X} \to \{0, 1\} | g(\mathbf{x}) = h(\mathbf{x}) \oplus h'(\mathbf{x}) \quad \forall h, h' \in \mathcal{H}\}$$
$\oplus$ *denotes the XOR operation.*

**Lemma 2 .**
$$\forall h, h' \in \mathcal{H}, |\epsilon_S(h, h') - \epsilon_T(h, h')| \leq \frac{1}{2} d_{\mathcal{H}\Delta\mathcal{H}}(\mathcal{D}_S, \mathcal{D}_T)$$

*Proof.* only need note that $h(\mathbf{x}) \oplus h'(\mathbf{x}) = |h(\mathbf{x}) - h'(\mathbf{x})|$, so
$$\sup_{g \in \mathcal{H}\Delta\mathcal{H}} |\Pr_{\mathcal{D}_S}[I(g)] - \Pr_{\mathcal{D}_T}[I(g)]| = \sup_{h, h' \in \mathcal{H}} |\epsilon_S(h, h') - \epsilon_T(h, h')|$$
this is done by definition. $\qquad\square$

With above notations we can derive a general proposition related with Theorem 3.

**Proposition 2 .** *Assumimg $\mathcal{H}$ is a hypothesis class from $\mathcal{X}$ to $\mathcal{Y}$ and $\phi, \phi' \in \mathcal{H}$, we have:*
$$\mathbb{E}_{\mathbf{x} \sim \mathcal{D}_T}\left[\ell(\phi(\mathbf{x}), \phi'(\mathbf{x}))\right] \leq \mathbb{E}_{\mathbf{x} \sim \mathcal{D}_S}\left[\ell(\phi(\mathbf{x}), \phi'(\mathbf{x}))\right] + D_{\mathrm{TV}}(\mathcal{D}_S, \mathcal{D}_T)$$
*where loss function is $\ell(\hat{y}, y) := \mathbf{1}(\hat{y} \neq y)$.*

*Proof.* using the lemmas above,
$$\epsilon_T(\phi, \phi') \leq \epsilon_S(\phi, \phi') + |\epsilon_T(\phi, \phi') - \epsilon_S(\phi, \phi')|$$
$$\leq \epsilon_S(\phi, \phi') + \frac{1}{2} d_{\mathcal{H}\Delta\mathcal{H}}(\mathcal{D}_S, \mathcal{D}_T)$$
$$\leq \epsilon_S(\phi, \phi') + D_{\mathrm{TV}}(\mathcal{D}_S, \mathcal{D}_T)$$

$\qquad\square$

To obtain some useful inequalities of generalization bound, we need to introduce the Rademacher complexity.

**Definition 9 (Rademacher complexity of a function class).** *Given a sample of points $S = \{z_1, z_2, \ldots, z_m\} \subset Z$, and considering a function class $\mathcal{F}$ of real-valued functions over Z, the empirical Rademacher complexity of $\mathcal{F}$ given S is defined as:*

$$\mathfrak{R}_S(\mathcal{F}) = \frac{1}{m}\mathbb{E}_\sigma\left[\sup_{f\in\mathcal{F}}\sum_{i=1}^m \sigma_i f(z_i)\right]$$

*where $\sigma_i$ are independent and identically distributed Rademacher random variables. In other words, for $i = 1, 2, \ldots, m$, the probability that $\sigma_i = +1$ is equal to the probability that $\sigma_i = -1$, and both are $\frac{1}{2}$. Further, let P be a probability distribution over Z. The Rademacher complexity of the function class $\mathcal{F}$ with respect to P for sample size m is:*

$$\mathfrak{R}_{P,m}(\mathcal{F}) := \mathbb{E}_{S\sim P^m}\left[\mathfrak{R}_S(\mathcal{F})\right]$$

**Lemma 3 (Generalization bound with Rademacher complexity).** *Let $\mathcal{F}$ be a family of loss functions $\mathcal{F} = \{(x, y) \mapsto \ell((x, y), h) : h \in \mathcal{H}\}$ with $\ell((x, y), h) \in [0, 1]$ for all $\ell, (x, y)$ and h. Then, with probability $1 - \delta$, the generalization gap is*

$$L(h) - \hat{L}(h) \leq 2\mathfrak{R}_U(\mathcal{F}) + 3\sqrt{\frac{\log(2/\delta)}{2n}} ,$$

*for all $h \in \mathcal{H}$ and samples U of size n.*

The proof of this classical result could be found in most machine learning textbooks, like [46]. Since $\ell(\cdot)$ is 1-Lipschitz, we can derive that $\mathfrak{R}_{U_S}(\ell \circ \mathcal{H}) \leq \mathfrak{R}_{U_S}(\mathcal{H})$.

Theorem 3 involves the representation distance, below we prove the triangle inequality for representation distance.

**Lemma 4 (Triangle inequality for Representation distance).** *for any functions $\phi_S, \phi_T, \phi_U$ and distribution D, we have:*

$$D_{\mathsf{Rep}}(\phi_S \to \phi_T; D) \leq D_{\mathsf{Rep}}(\phi_S \to \phi_U; D) + D_{\mathsf{Rep}}(\phi_U \to \phi_T; D)$$

*Proof.* Let's denote:
$$d_1 = D_{\mathsf{Rep}}(\phi_S \to \phi_U; D)$$
$$d_2 = D_{\mathsf{Rep}}(\phi_U \to \phi_T; D)$$

By definition:
$$d_1 = \inf_{W_1\in\mathbb{R}^{m\times n}, b_1\in\mathbb{R}^m} \mathbb{E}_{x\sim D}\ell(W_1\phi_S(x) + b_1, \phi_U(x))$$
$$d_2 = \inf_{W_2\in\mathbb{R}^{m\times n}, b_2\in\mathbb{R}^m} \mathbb{E}_{x\sim D}\ell(W_2\phi_U(x) + b_2, \phi_T(x))$$

We need to show:
$$D_{\mathsf{Rep}}(\phi_S \to \phi_T; D) \leq d_1 + d_2$$

Consider the composition of the transformations:
$$\phi_S(x) \xrightarrow{W_1, b_1} \phi_U(x) \xrightarrow{W_2, b_2} \phi_T(x)$$

The combined transformation can be written as:
$$W_2(W_1\phi_S(x) + b_1) + b_2 = W_2 W_1 \phi_S(x) + W_2 b_1 + b_2$$

Using the properties of the loss function $\ell$, we can write:

$$\ell(W_2 W_1 \phi_S(x) + W_2 b_1 + b_2, \phi_T(x)) \leq \ell(W_1\phi_S(x) + b_1, \phi_U(x)) + \ell(W_2\phi_U(x) + b_2, \phi_T(x))$$

This inequality holds for the reason that it can only break when the two items at right-hand side are both 0, in which the left side is also 0 due to rule of composition. Thus the inequality holds for all $x$ and $W_1, b_1, W_2, b_2$.

Taking expectations over $x \sim D$:

$$\mathbb{E}_{x \sim D} \ell(W_2 W_1 \phi_S(x) + W_2 b_1 + b_2, \phi_T(x)) \leq \mathbb{E}_{x \sim D} \ell(W_1 \phi_S(x) + b_1, \phi_U(x)) + \mathbb{E}_{x \sim D} \ell(W_2 \phi_U(x) + b_2, \phi_T(x))$$

Taking the infimum over $W_1, b_1$ and $W_2, b_2$:

$$\inf_{W_1, b_1, W_2, b_2} \mathbb{E}_{x \sim D} \ell(W_2 W_1 \phi_S(x) + W_2 b_1 + b_2, \phi_T(x)) \leq d_1 + d_2$$

Since the left-hand side is an upper bound for $D_{\mathsf{Rep}}(\phi_S \to \phi_T; D)$, we have:

$$D_{\mathsf{Rep}}(\phi_S \to \phi_T; D) \leq d_1 + d_2$$

Therefore, the triangle inequality holds for the metric $D_{\mathsf{Rep}}$. $\qquad\square$

Now we are ready to prove Theorem 3.

*Proof.* For arbitrary $W, b$, with probability $1 - \delta$:

$$D_{\mathsf{Rep}}(\phi_{\boldsymbol{\theta}} \to \phi^\star; X) \leq \epsilon_X(W \phi_{\boldsymbol{\theta}} + b, \phi^\star) \leq \epsilon_{D_X}(W \phi_{\boldsymbol{\theta}} + b, \phi^\star) + 2\mathfrak{R}_{D_X}(\Phi) + 3\sqrt{\frac{\log(2/\delta)}{2|D_X|}}$$

Taking the infimum over $W, b$:

$$D_{\mathsf{Rep}}(\phi_{\boldsymbol{\theta}} \to \phi^\star; X) \leq D_{\mathsf{Rep}}(\phi_{\boldsymbol{\theta}} \to \phi^\star; D_X) + 2\mathfrak{R}_{D_X}(\Phi) + 3\sqrt{\frac{\log(2/\delta)}{2|D_X|}}$$

Using the triangle inequality for representation distance:

$$D_{\mathsf{Rep}}(\phi_{\boldsymbol{\theta}} \to \phi^\star; D_X) \leq D_{\mathsf{Rep}}(\phi_{\boldsymbol{\theta}} \to \psi; D_X) + D_{\mathsf{Rep}}(\psi \to \phi^\star; D_X)$$

For arbitrary $W, b$, using the proposition above,

$$D_{\mathsf{Rep}}(\phi_{\boldsymbol{\theta}} \to \psi; D_X) \leq \epsilon_{D_X}(W \phi_{\boldsymbol{\theta}} + b, \psi) \leq \epsilon_{S_X}(W \phi_{\boldsymbol{\theta}} + b, \psi) + D_{\mathsf{TV}}(S_X, D_X)$$

Taking the infimum over $W, b$:

$$D_{\mathsf{Rep}}(\phi_{\boldsymbol{\theta}} \to \psi; D_X) \leq D_{\mathsf{Rep}}(\phi_{\boldsymbol{\theta}} \to \psi; S_X) + D_{\mathsf{TV}}(S_X, D_X)$$

Combining the above results, we have:

$$D_{\mathsf{Rep}}(\phi_{\boldsymbol{\theta}} \to \phi^\star; X) \leq D_{\mathsf{Rep}}(\psi \to \phi^\star; D_X) + D_{\mathsf{Rep}}(\phi_{\boldsymbol{\theta}} \to \psi; S_X) + 2\mathfrak{R}_{D_X}(\Phi) + 3\sqrt{\frac{\log(2/\delta)}{2|D_X|}}$$

$$+ D_{\mathsf{TV}}(S_X, D_X)$$

$\qquad\square$

**Impact of initialization** In this part we leverage the Neural Tangent Kernel (NTK) framework [29] to deduce that a neural network initialized closer to a target function $f^\star$ converges faster during training. Under the NTK regime, neural networks exhibit linearized training dynamics, allowing us to predict how the network's output evolves during training.

In the context of supervised learning, consider a neural network with parameters $\theta$ and output function $f(x; \theta)$, where $x$ is the input. The goal is to approximate a target function $f^\star(x)$ by minimizing a loss function, typically the mean squared error (MSE):

$$L(\theta) = \frac{1}{2} \sum_{i=1}^{n} (f(x_i; \theta) - y_i)^2$$

where $\{(x_i, y_i)\}_{i=1}^n$ is the training data and $y_i = f^\star(x_i)$.

Under gradient descent with learning rate $\eta$, the parameter updates are:

$$\theta_{t+1} = \theta_t - \eta \nabla_\theta L(\theta_t)$$

In the NTK regime, where the network width tends to infinity, the network's output evolves linearly with respect to the parameters around initialization $\theta_0$ [38]. We can approximate:

$$f(x; \theta) \approx f(x; \theta_0) + \nabla_\theta f(x; \theta_0)^\top (\theta - \theta_0)$$

This linearization allows us to express the evolution of the network's output as:

$$f_t(x) = f_0(x) - \eta \sum_{s=0}^{t-1} \sum_{i=1}^n K(x, x_i)(f_s(x_i) - y_i)$$

where $f_t(x) = f(x; \theta_t)$ is the network output at time $t$ and $K(x, x') = \nabla_\theta f(x; \theta_0)^\top \nabla_\theta f(x'; \theta_0)$ is the Neural Tangent Kernel.

In continuous time (gradient flow), the training dynamics can be described by a differential equation:

$$\frac{df_t(x)}{dt} = -\sum_{i=1}^n K(x, x_i)(f_t(x_i) - y_i)$$

Vectorizing over all training inputs, we have:

$$\frac{d\mathbf{f}_t}{dt} = -\mathbf{K}(\mathbf{f}_t - \mathbf{y})$$

where $\mathbf{f}_t = [f_t(x_1), f_t(x_2), \ldots, f_t(x_n)]^\top$, $\mathbf{y} = [y_1, y_2, \ldots, y_n]^\top$ and $\mathbf{K}$ is the NTK matrix with entries $K_{ij} = K(x_i, x_j)$. This differential equation has the solution:

$$\mathbf{f}_t - \mathbf{y} = e^{-\mathbf{K}t}(\mathbf{f}_0 - \mathbf{y})$$

This equation shows that the error $\mathbf{f}_t - \mathbf{y}$ decays exponentially over time, with the rate governed by the NTK matrix $\mathbf{K}$. Since $\mathbf{K}$ is symmetric, we can decompose $\mathbf{K}$ into its eigenvalues $\{\lambda_j\}$ and corresponding eigenvectors $\{\mathbf{v}_j\}$:

$$\mathbf{K} = \sum_j \lambda_j \mathbf{v}_j \mathbf{v}_j^\top$$

the solution becomes:

$$\mathbf{f}_t - \mathbf{y} = \sum_j e^{-\lambda_j t} c_j \mathbf{v}_j$$

where $c_j = \mathbf{v}_j^\top (\mathbf{f}_0 - \mathbf{y})$. The rate of convergence for each component of the error is proportional to the corresponding eigenvalue $\lambda_j$ of the NTK. Larger eigenvalues lead to faster decay. Initial error matters: the initial error $\mathbf{f}_0 - \mathbf{y}$ scales the amplitude of the exponential terms. A smaller initial error means the network starts closer to the target function, reducing the time required for the error to decay to a specific threshold.

Suppose we want the error to be less than $\epsilon$:

$$\|\mathbf{f}_t - \mathbf{y}\| \leq \epsilon$$

Using the solution:

$$\|\mathbf{f}_t - \mathbf{y}\| \leq \|e^{-\lambda_{\min}t}\|\|\mathbf{f}_0 - \mathbf{y}\|$$

where $\lambda_{\min}$ is the smallest (positive) eigenvalue of $\mathbf{K}$. Solving for time $t$:

$$t \geq \frac{1}{\lambda_{\min}} \ln\left(\frac{\|\mathbf{f}_0 - \mathbf{y}\|}{\epsilon}\right)$$

By leveraging the NTK framework, we've shown that a better initialization(meaning the neural network's initial output function is closer to the target function $f^\star$) leads to faster convergence during training. This is because the training dynamics under NTK are linear, and the exponential error decay is directly influenced by the magnitude of the initial error. Analysis above implies that if we can train a network that is closer to the target function, it will converge faster to the target function.

### D.1 Relation between data distribution and Rademacher complexity

Now let's look at the Rademacher complexity appeared above more carefully. Let 1-Lipschitz positive homogeneous activation $\sigma_i$ be given, and

$$\mathcal{H} := \left\{ \mathbf{x} \mapsto \sigma_L\left(W_L\sigma_{L-1}\left(\cdots\sigma_1\left(W_1\mathbf{x}\right)\cdots\right)\right) : \|W_i\|_{\mathrm{F}} \leq B, \mathbf{x} \in \mathbb{R}^d \right\}.$$

Then using Theorem 1 in [23], for samples $S$ of size $m$ we have bound for the empirical Rademacher complexity:

$$\mathfrak{R}_S(\mathcal{H}) \leq \frac{1 + \sqrt{2L\ln 2}}{m} B^L \|X\|_{\mathrm{F}}$$

where $X \in \mathbb{R}^{d\times m}$ is the input data matrix, and $\|\cdot\|_{\mathrm{F}}$ denotes the Frobenius norm.

If we further assume that the data are drawn from the distribution $G$ (stated in Section 3.2) with covariance $\Sigma$ (for simplicity let $\mu_1 = -\mu, \mu_2 = \mu$), then we can bound the Rademacher complexity of $\mathcal{H}$ with respect to $G$:

$$
\begin{aligned}
\mathfrak{R}_{G,m}(\mathcal{H}) &= \mathbb{E}_{S\sim G^m}\left[\mathfrak{R}_S(\mathcal{H})\right] \\
&\leq \frac{1 + \sqrt{2L\ln 2}}{m} B^L\, \mathbb{E}_{S\sim G^m}\|X\|_{\mathrm{F}} \\
&= \frac{\sqrt{2} + 2\sqrt{L\ln 2}}{m} B^L \cdot \Sigma\, \Gamma\left(\frac{1+dm}{2}\right) M\left(-\frac{1}{2}, \frac{dm}{2}, -\frac{dm\mu^2}{2\Sigma^2}\right)/\Gamma\left(\frac{dm}{2}\right)
\end{aligned}
$$

The right part of the last inequality is an increasing function with respect to $\Sigma$, and $\Gamma(\cdot)$ denotes the gamma function, $M(\cdot, \cdot, \cdot)$ is the Kummer's confluent hypergeometric function, given by:

$$M(a, b, z) = \sum_{n=0}^{\infty} \frac{a^{(n)} z^n}{b^{(n)} n!} = {}_1F_1(a; b; z),$$

where:

$$a^{(0)} = 1,$$
$$a^{(n)} = a(a+1)(a+2)\cdots(a+n-1),$$

is the rising factorial.

**Remark.** Usually, the Rademacher complexity $\mathfrak{R}_S(\ell \circ \mathcal{F})$ could be bounded by of $\mathfrak{R}_S(\mathcal{F})$. For example, if $\ell$ is $L$-lipschitz, then $\mathfrak{R}_S(\ell \circ \mathcal{F}) \leq L \cdot \mathfrak{R}_S(\mathcal{F})$. That's why we directly compute the Rademacher complexity of $\mathcal{H}$ instead of $\ell \circ \mathcal{H}$.

## E   Explanation of Rescaling Samples

Dataset distillation seeks to create a condensed dataset that allows models to achieve performance comparable to those trained on the full dataset, but with fewer training steps. In this section, we will demonstrate that rescaling the variance of Gaussian distributions, as defined in Definition 3, does not affect the optimal performance of models trained on these rescaled data.

*Proof.* Consider two Gaussian distributions $\mathcal{N}_0(\mu_1, \sigma^2 \mathbf{I})$ and $\mathcal{N}_1(\mu_2, \sigma^2 \mathbf{I})$ with means $\mu_1 = 1$ and $\mu_2 = 2$, and variance $\sigma^2$. We define the bimodal mixture distribution $G = (G_X, G_Y)$ such that $(\mathbf{x}, y)$ is sampled according to:

$$\mathbf{x} = (1 - y) \cdot \mathbf{x}_0 + y \cdot \mathbf{x}_1, \quad \text{with} \quad y \sim \text{Bernoulli}(0.5), \quad \mathbf{x}_0 \sim \mathcal{N}_0, \quad \mathbf{x}_1 \sim \mathcal{N}_1.$$

The decision rule for optimal classification is determined by the likelihood ratio test. For a given sample $\mathbf{x}$, the log-likelihood ratio $\Lambda(\mathbf{x})$ is given by:

$$\Lambda(\mathbf{x}) = \log \frac{P(\mathbf{x} \mid y = 1)}{P(\mathbf{x} \mid y = 0)}.$$

Since $\mathbf{x}_0$ and $\mathbf{x}_1$ are drawn from Gaussian distributions, their probability density functions are:

$$P(\mathbf{x} \mid y = 0) = \frac{1}{(2\pi\sigma^2)^{d/2}} \exp\left(-\frac{1}{2\sigma^2}\|\mathbf{x} - \mu_1\|^2\right),$$

$$P(\mathbf{x} \mid y = 1) = \frac{1}{(2\pi\sigma^2)^{d/2}} \exp\left(-\frac{1}{2\sigma^2}\|\mathbf{x} - \mu_2\|^2\right).$$

Substituting these into the log-likelihood ratio, we have:

$$\Lambda(\mathbf{x}) = \log \frac{\frac{1}{(2\pi\sigma^2)^{d/2}} \exp\left(-\frac{1}{2\sigma^2}\|\mathbf{x} - \mu_2\|^2\right)}{\frac{1}{(2\pi\sigma^2)^{d/2}} \exp\left(-\frac{1}{2\sigma^2}\|\mathbf{x} - \mu_1\|^2\right)}.$$

Simplifying, we obtain:

$$\Lambda(\mathbf{x}) = -\frac{1}{2\sigma^2}\|\mathbf{x} - \mu_2\|^2 + \frac{1}{2\sigma^2}\|\mathbf{x} - \mu_1\|^2.$$

Further simplification gives:

$$\Lambda(\mathbf{x}) = \frac{1}{2\sigma^2}\left(\|\mathbf{x} - \mu_1\|^2 - \|\mathbf{x} - \mu_2\|^2\right).$$

Since the optimal decision threshold for balanced classes (i.e., $P(y = 1) = P(y = 0) = 0.5$) is $\Lambda(\mathbf{x}) = 0$, we set:

$$\|\mathbf{x} - \mu_1\|^2 - \|\mathbf{x} - \mu_2\|^2 = 0.$$

Expanding and rearranging, we derive:

$$\|\mathbf{x}\|^2 - 2\mu_1\mathbf{x} + \mu_1^2 - \|\mathbf{x}\|^2 + 2\mu_2\mathbf{x} - \mu_2^2 = 0.$$

This simplifies to:

$$2(\mu_2 - \mu_1)\mathbf{x} + (\mu_1^2 - \mu_2^2) = 0,$$

$$2(\mu_2 - \mu_1)\mathbf{x} = \mu_2^2 - \mu_1^2,$$

$$\mathbf{x} = \frac{\mu_2^2 - \mu_1^2}{2(\mu_2 - \mu_1)}.$$

Solving yields:

$$\mathbf{x} = \frac{\mu_2 + \mu_1}{2}.$$

Therefore, the optimal decision boundary is $x = 1.5$, which is independent of the variance $\sigma^2$. This completes the proof. $\square$

Regarding efficiency, utilizing scaled data to train similar models with fewer training steps is proven in Section B. Additionally, rescaling each Gaussian distribution preserves their means, which aligns with the objectives of conventional distribution matching-based dataset distillation methods. These methods aim to distill data while maintaining the distributional properties, specifically their means.

# F  Detailed Methodology of RELA-D

Recall that $\phi_{\boldsymbol{\theta}} : \mathbb{R}^d \to \mathbb{R}^n$. We then introduce a transport matrix $\mathbf{W} \in \mathbb{R}^{m \times n}$ and define the combined model as $\mathbf{W}\phi_{\boldsymbol{\theta}}(\cdot) \in \mathbb{R}^m$. During the training phase, the parameters $\mathbf{W}$ and $\boldsymbol{\theta}$ are jointly optimized. However, as the dimension $m$ increases, the computational complexity of the optimization grows rapidly. To address this issue, we propose reducing the dimensionality of the target matrix $R_{\mathbf{Y}}$ from $m$ to $n$, where $n < m$:

$$R'_{\mathbf{Y}} = V_n^\top \left( \frac{R_{\mathbf{Y}} - \mu}{\sigma} \right) \quad \text{s.t.} \quad \frac{1}{|R_{\mathbf{Y}}| - 1} \left( (R_{\mathbf{Y}} - \mu)^\top (R_{\mathbf{Y}} - \mu) \right) = V \Lambda V^\top, \ V_n = V[:, :n], \qquad (10)$$

where $\mu$ and $\sigma$ denote the mean and standard deviation, respectively, of each column in the $R_{\mathbf{Y}}$. Practically, we use batch PCA to perform the computation shown in (10), as illustrated in L.

## F.1  Proof for Ideal Properties of Prior Models

We aim to demonstrate that modern deep learning methods can effectively train models to serve as robust prior models by extracting sufficient information from samples as representations.

Therefore, we poist the existence of a prior model $\xi$ capable of losslessly extracting the information of samples $D_{\mathcal{X}}$ when trained using the InfoNCE loss [47], a method prevalently employed in contemporary deep learning algorithms [12].

*Proof.* To demonstrate that an encoder $\xi$ trained with the InfoNCE loss preserves all information from the input data $D_{\mathcal{X}}$, we proceed as follows.

**1. Definitions and Setup**

Let $\mathcal{X}$ denote the input data space with data distribution $p_{\text{data}}(x)$. Let $p_{\text{pos}}(x, x^+)$ denote the distribution of positive pairs, typically generated via data augmentation. The encoder $\xi : \mathcal{X} \to \mathbb{R}^d$ maps inputs to $d$-dimensional representations. The similarity function $q : \mathbb{R}^d \times \mathbb{R}^d \to \mathbb{R}$ (e.g., dot product) measures similarity between representations. Given a positive pair $(x, x^+)$, let $\{x_i^-\}_{i=1}^N$ be $N$ negative samples drawn i.i.d. from $p_{\text{data}}(x)$. The InfoNCE loss is defined as:

$$\mathcal{L}_{\text{InfoNCE}}(\xi, q) = -\mathbb{E}_{(x, x^+) \sim p_{\text{pos}}} \left[ \log \frac{e^{q(\xi(x), \xi(x^+))/\tau}}{\sum_{i=1}^N e^{q(\xi(x), \xi(x_i^-))/\tau}} \right]$$

where $\tau > 0$ is a temperature parameter.

**2. InfoNCE as a Mutual Information Lower Bound**

The InfoNCE objective serves as a lower bound to the mutual information between representations of positive pairs:

$$I(\xi(X); \xi(X^+)) \geq \mathcal{J}(\xi, q)$$

where

$$\mathcal{J}(\xi, q) = \mathbb{E}_{(x, x^+) \sim p_{\text{pos}}} \left[ q(\xi(x), \xi(x^+)) - \tau \cdot \log \left( \sum_{i=1}^N e^{q(\xi(x), \xi(x_i^-))/\tau} \right) \right]$$

As the number of negative samples $N \to \infty$, this bound becomes tight, approaching the true mutual information $I(\xi(X); \xi(X^+))$. **3. Optimal Encoder and Similarity Function** Let $(\xi^*, q^*)$ denote the optimal encoder and similarity function that maximize $\mathcal{J}(\xi, q)$. Under the assumption of infinite negative samples and an expressive similarity function, the optimal similarity satisfies:

$$q^*(\xi^*(x), \xi^*(x^+)) = \log \frac{p_{\text{pos}}(x, x^+)}{p_{\text{data}}(x) p_{\text{data}}(x^+)} + C$$

where $C$ is a constant independent of $(x, x^+)$. This aligns $q^*$ with the pointwise mutual information (PMI) between $x$ and $x^+$. **4. Injectivity through Mutual Information Maximization** Maximizing $I(\xi(X); \xi(X^+))$ encourages the encoder $\xi$ to capture as much information about $X$ as possible. To ensure injectivity:

- **Sufficient Dimensionality:** The representation dimension $d$ must be at least as large as the intrinsic dimensionality of $\mathcal{X}$.

Table 4: Notations used in this section.

| Notation | Meaning |
|---|---|
| $u_1, u_2$ | current concerned samples |
| $v$ | unspecified samples |
| $u_1^o, v^o$ | samples from online branch |
| $u_2^t, v^t$ | samples from unspecified target branch |
| $u_2^s, v^s$ | samples from weight-sharing target branch |
| $u_2^d, v^d$ | samples from stop-gradient target branch |
| $u_2^m, v^m$ | samples from momentum-encoder target branch |
| $\mathcal{V}$ | unspecified sample set |
| $\mathcal{V}_{\text{batch}}$ | sample set of current batch |
| $\mathcal{V}_{\text{bank}}$ | sample set of memory bank |
| $\mathcal{V}_{\infty}$ | sample set of all previous samples |

- **Expressive Architecture:** The encoder $\xi$ should be sufficiently expressive, potentially utilizing architectural constraints (e.g., invertible networks) to promote injectivity.

Under these conditions, maximizing mutual information implies that $\xi^*$ approximates an injective mapping on the support of $p_{\text{data}}(x)$, i.e., $\xi^*(x) = \xi^*(x') \Rightarrow x = x'$ almost surely. **5. Existence of the Inverse Mapping** Given that $\xi^*$ is injective, there exists a deterministic inverse mapping $g : \mathbb{R}^d \to \mathcal{X}$ such that:

$$g(\xi^*(x)) = x \quad \text{for all } x \in \mathcal{X}$$

This mapping $g$ can be constructed as the inverse of $\xi^*$ on its image:

$$g(z) = \xi^{*-1}(z) \quad \text{where } z \in \xi^*(\mathcal{X})$$

$\square$

# G   Analysis of Different Self-Supervised Learning Methods

We refer to the primary theoretical results for existing self-supervised learning methods as presented in [59], where all methods are unified under a simple and cohesive framework, detailed below. Specifically, UniGrad [59] demonstrates that most self-supervised learning methods can be unified by analyzing their gradients.

## G.1   A Unified Framework for SSL

A typical self-supervised learning framework employs a siamese network with two branches: an online branch and a target branch. The target branch serves as the training target for the online branch.

Given an input image $x$, two augmented views $x_1$ and $x_2$ are generated, serving as inputs for the two branches. The encoder $f(\cdot)$ extracts representations $u_i \triangleq f(x_i)$ for $i = 1, 2$ from these views.

Table 4 details the notations used. $u_1$ and $u_2$ denote the current training samples, while $v$ denotes unspecified samples. $u_1^o$ and $v^o$ are representations from the online branch. Three types of target branches are widely used: 1) weight-sharing with the online branch ($u_2^s$ and $v^s$); 2) weight-sharing but detached from gradient back-propagation ($u_2^d$ and $v^d$); 3) a momentum encoder updated from the online branch ($u_2^m$ and $v^m$). If unspecified, $u_2^t$ and $v^t$ are used. A symmetric loss is applied to the two augmented views, as described in [14].

$\mathcal{V}$ represents the sample set in the current training step. Methods vary in constructing this set: $\mathcal{V}_{\text{batch}}$ includes all samples from the current batch, $\mathcal{V}_{\text{bank}}$ uses a memory bank storing previous samples, and $\mathcal{V}_{\infty}$ includes all previous samples, potentially larger than a memory bank.

## G.2   Contrastive Learning Methods

Contrastive learning relies on negative samples to prevent representational collapse and enhance performance. Positive samples are derived from different views of the same image, while negative

samples come from other images. The goal is to attract positive pairs and repel negative pairs, typically using the InfoNCE loss [47]:

$$L = \mathop{\mathbb{E}}_{u_1, u_2} \left[ -\log \frac{\exp\left(\cos(u_1^o, u_2^t)/\tau\right)}{\sum_{v^t \in \mathcal{V}} \exp\left(\cos(u_1^o, v^t)/\tau\right)} \right], \tag{11}$$

where $\cos(\cdot)$ denotes cosine similarity, and $\tau$ is the temperature hyper-parameter. This formulation can be adapted for various methods, discussed below.

**MoCo [25, 13].** MoCo uses a momentum encoder for the target branch and a memory bank for storing previous representations. Negative samples are drawn from this memory bank. The gradient for sample $u_1^o$ is:

$$\frac{\partial L}{\partial u_1^o} = \frac{1}{\tau N} \left( -u_2^m + \sum_{v^m \in \mathcal{V}_{\text{bank}}} s_v v^m \right), \tag{12}$$

where $s_v = \frac{\exp\left(\cos(u_1^o, v^m)/\tau\right)}{\sum_{y^m \in \mathcal{V}_{\text{bank}}} \exp\left(\cos(u_1^o, y^m)/\tau\right)}$ and $N$ is the number of samples in the batch.

**SimCLR [12].** SimCLR shares weights between the target and online branches and does not stop back-propagation. It uses all representations from other images in the batch as negative samples. The gradient is:

$$\begin{aligned} \frac{\partial L}{\partial u_1^o} = & \frac{1}{\tau N} \left( -u_2^s + \sum_{v^s \in \mathcal{V}_{\text{batch}} \setminus u_1^o} s_v v^s \right) \\ & + \underbrace{\frac{1}{\tau N} \left( -u_2^s + \sum_{v^s \in \mathcal{V}_{\text{batch}} \setminus u_1^o} t_v v^s \right)}_{\text{reduce to } 0}, \end{aligned} \tag{13}$$

where $t_v = \frac{\exp\left(\cos(v^s, u_1^o)/\tau\right)}{\sum_{y^s \in \mathcal{V}_{\text{batch}} \setminus v^s} \exp\left(\cos(v^s, y^s)/\tau\right)}$. If the gradient through the target branch is stopped, the second term vanishes.

**Unified Gradient.** The gradient for these methods can be unified as:

$$\frac{\partial L}{\partial u_1^o} = \frac{1}{\tau N} \left( -u_2^t + \sum_{v^t \in \mathcal{V}} s_v v^t \right), \tag{14}$$

comprising a weighted sum of positive and negative samples. The term $-u_2^t$ pulls positive samples together, while $\sum_{v^t \in \mathcal{V}} s_v v^t$ pushes negative samples apart. The main difference between methods lies in the target branch used and the construction of the contrastive sample set $\mathcal{V}$.

### G.3 Asymmetric Network Methods

Asymmetric network methods learn representations by maximizing the similarity of positive pairs without using negative samples. These methods require symmetry-breaking network designs to avoid representational collapse. A predictor $h(\cdot)$ is appended after the online branch, and the gradient to the target branch is stopped. The objective function is:

$$L = \mathop{\mathbb{E}}_{u_1, u_2} \left[ -\cos(h(u_1^o), u_2^t) \right]. \tag{15}$$

**Relation to BYOL [30].** BYOL uses a momentum encoder for the target branch, i.e., $u_2^t = u_2^m$ in Eq.(15).

**Relation to Simsiam [14].** Simsiam shows that a momentum encoder is unnecessary and only applies the stop-gradient operation to the target branch, i.e., $u_2^t = u_2^d$ in Eq.(15).

**Unified Gradient.** Despite the performance of asymmetric network methods, the avoidance of collapse solutions is not well understood. DirectPred [60] explores this by studying training dynamics and proposes an analytical solution for the predictor $h(\cdot)$.

DirectPred formulates the predictor as $h(v) = W_h v$, with $W_h$ calculated based on the correlation matrix $\mathbb{E}_v(vv^T)$. The correlation matrix, $F$, is computed as the moving average for each batch:

$F \triangleq \sum_{v^o \in \mathcal{V}_\infty} \rho_v v^o v^{oT}$, where $\rho_v$ is the moving average weight. Decomposing $F$ into eigenvalues $\Lambda_F$ and eigenvectors $U$, $W_h$ is:

$$W_h = U\Lambda_h U^T, \ \ \Lambda_h = \Lambda_F^{1/2} + \epsilon\lambda_{max}I, \tag{16}$$

where $\lambda_{max}$ is the max eigenvalue of $F$ and $\epsilon$ is a hyper-parameter to boost small eigenvalues.

DirectPred also derives the gradient:

$$\frac{\partial L}{\partial u_1^o} = \frac{1}{||W_h u_1^o||_2 N}\left(-W_h^T u_2^t + \lambda \sum_{v^o \in \mathcal{V}_\infty}(\rho_v u_1^{oT}v^o)v^o\right), \tag{17}$$

where $-W_h^T u_2^t$ and $\sum_{v^o \in \mathcal{V}_\infty}(\rho_v u_1^{oT}v^o)v^o$ act as positive and negative gradients respectively, and $\lambda = \frac{u_1^{oT}W_h^T u_2^t}{u_1^{oT}(F+\epsilon^2 I)u_1^o}$ is a balance factor.

Though negative samples are absent in the loss function, they emerge from the predictor network's optimization. The eigenspace of the predictor $W_h$ aligns with the feature correlation matrix $F$, encoding its information. During back-propagation, this encoded information functions as a negative gradient, influencing the optimization direction.

### G.4 Feature Decorrelation Methods

Feature decorrelation methods have recently emerged as a novel approach in self-supervised learning. These methods aim to reduce redundancy among different feature dimensions to prevent collapse. Various loss functions have been proposed for this purpose. We examine their relations below.

**Relation to Barlow Twins [69]**. Barlow Twins employs the following loss function:

$$L = \sum_{i=1}^C (W_{ii} - 1)^2 + \lambda \sum_{i=1}^C \sum_{j \neq i} W_{ij}^2, \tag{18}$$

where $W = \frac{1}{N}\sum_{v_1^o, v_2^s \in \mathcal{V}_{\text{batch}}} v_1^o v_2^{sT}$ is a cross-correlation matrix, $C$ denotes the number of feature dimensions, and $\lambda$ is a balancing hyper-parameter. The diagonal elements of $W$ are encouraged to be close to 1, while the off-diagonal elements are forced towards 0.

Despite appearing different, Eq. (18) operates similarly to previous methods from a gradient perspective, calculated as:

$$\frac{\partial L}{\partial u_1^o} = \frac{2}{N}\left(-Au_2^s + \lambda \sum_{v_1^o, v_2^s \in \mathcal{V}_{\text{batch}}} \frac{u_2^{sT}v_2^s}{N}v_1^o\right), \tag{19}$$

where $A = I - (1-\lambda)W_{\text{diag}}$ and $(W_{\text{diag}})_{ij} = \delta_{ij}W_{ij}$ is the diagonal matrix of $W$. Barlow Twins applies batch normalization instead of $\ell_2$ normalization to the representation $v$.

**Relation to VICReg [3]**. VICReg modifies Barlow Twins with the following loss function:

$$\begin{aligned}
L = &\frac{1}{N}\sum_{v_1^o, v_2^s \in \mathcal{V}_{\text{batch}}} ||v_1^o - v_2^s||_2^2 + \frac{\lambda_1}{C}\sum_{i=1}^C \sum_{j \neq i} W_{ij}'^2 \\
&+ \frac{\lambda_2}{C}\sum_{i=1}^C \max(0, \gamma - \text{std}(v_1^o)_i),
\end{aligned} \tag{20}$$

where $W' = \frac{1}{N-1}\sum_{v_1^o \in \mathcal{V}_{\text{batch}}}(v_1^o - \bar{v}_1^o)(v_1^o - \bar{v}_1^o)^T$ is the covariance matrix of the same view, $\text{std}(v)_i$ denotes the standard deviation of the $i$-th channel of $v$, $\gamma$ is a constant target value, and $\lambda_1, \lambda_2$ are balancing weights.

The gradient is derived as follows:

$$\begin{aligned}
\frac{\partial L}{\partial u_1^o} = &\frac{2}{N}\left(-u_2^s + \lambda \sum_{v_1^o \in \mathcal{V}_{\text{batch}}} \frac{\tilde{u}_1^{oT}\tilde{v}_1^o}{N}\tilde{v}_1^o\right) \\
&+ \underbrace{\frac{2\lambda}{N}\left(\frac{1}{\lambda}u_1^o - B\tilde{u}_1^o\right)}_{\text{reduces to 0}},
\end{aligned} \tag{21}$$

where $\tilde{v} = v - \bar{v}$ is the de-centered sample, $\lambda = \frac{2\lambda_1 N^2}{c(N-1)^2}$, and $B = \frac{N}{c\lambda(N-1)}(2\lambda_1 W'_{\text{diag}} + \frac{\lambda_2}{2}\text{diag}(\mathbf{1}(\gamma - \text{std}(v_1^o) > 0) \oslash \text{std}(v_1^o)))$. Here, $\text{diag}(x)$ is a matrix with the vector $x$ on its diagonal, $\mathbf{1}(\cdot)$ is the indicator function, and $\oslash$ denotes element-wise division.

VICReg does not normalize $v$; instead, it uses de-centering and a standard deviation term in the loss function.

**Unified Gradient.** Given the equivalence of $v^s$ and $v^o$, the gradient for feature decorrelation methods can be unified as:

$$\frac{\partial L}{\partial u_1^o} = \frac{2}{N}\left(-u_2^t + \lambda \sum_{v^o \in \mathcal{V}_{\text{batch}}} \frac{u^{oT}v^o}{N}v_1^o\right), \tag{22}$$

where $-u_2^t$ is the positive gradient, and $\sum_{v^o \in \mathcal{V}_{\text{batch}}}\left(\frac{u^{oT}v^o}{N}\right)v_1^o$ is the negative gradient. $\lambda$ is a balancing factor. The difference between methods lies in the subscript for the negative coefficient. Feature decorrelation methods function similarly to other self-supervised methods, with the positive and negative gradients derived from the diagonal and off-diagonal elements of the correlation matrix.

# H   Budget of RELA for Data Synthesis

While training on the distilled dataset is both efficient and effective, the distillation process in optimization-based approaches [11, 72, 65] is computationally intensive [18, 56], often exceeding the computational load of training on the full dataset.

In contrast, the synthetic data generated by our RELA framework requires a budget less than that of training for a single epoch (c.f. Section 4.1). This is because the synthesis budget of our RELA is equivalent to performing inference over the entire dataset $D_X$. Consequently, this computational expense is negligible given that training epochs typically number around 100.

# I   RELA in Labeled Dataset Distillation and Human-Supervised Learning

We apply our RELA in labeled dataset distillation and human-supervised learning.

## I.1   Experimental Setup

**Datasets and Neural Network Architectures.** We conduct experiments on datasets of varying scales and resolutions.

- **Small-scale:** We evaluate on CIFAR-10 ($32 \times 32$) [35] and CIFAR-100 ($32 \times 32$) [34].
- **Large-scale:** We utilize Tiny-ImageNet ($64 \times 64$) [36] and ImageNet-1K ($224 \times 224$) [20].

Consistent with previous works on dataset distillation [67, 73, 24], we use ConvNet [24] and ResNet-{18,50} [26] as our backbone networks across all datasets. Specifically, Conv-3 is employed for CIFAR-10/100, and Conv-4 for Tiny-ImageNet and ImageNet-1K.

**Baselines.** We compare our method with several SOTA distillation methods capable of scaling to large high-resolution datasets, including G-VBSM [53], SRe$^2$L [67], and RDED [56]. To the best of our knowledge, SRe$^2$L, G-VBSM, and RDED are the only published works that efficiently scale to datasets of any size, making them our closest baselines. All distilled datasets synthesized from these baselines undergo the same post-training process. Results are reported in Table 5. For our RELA, the prior models used for distillation are identical to the pre-trained models employed in the baseline methods.

## I.2   Main Results

Table 5 demonstrates the superiority of our RELA, despite incorporating a zero-cost sample synthesis process.

Table 5: **Comparison with SOTA Baseline Dataset Distillation Methods.** In the table, **bold** indicates the best result. IPC refers to the number of Images Per Class for distilled datasets.

| Dataset | IPC | ConvNet | | | | ResNet-18 | | | | ResNet-50 | | | |
|---|---|---|---|---|---|---|---|---|---|---|---|---|---|
| | | G-VBSM | SRe2L | RDED | RELA (Ours) | G-VBSM | SRe2L | RDED | RELA (Ours) | G-VBSM | SRe2L | RDED | RELA (Ours) |
| CF-10 | 1 | 21.2 ± 0.2 | 22.2 ± 1.1 | 28.9 ± 0.4 | **40.3 ± 0.4** | 17.0 ± 1.0 | 19.9 ± 0.9 | 27.8 ± 0.4 | **35.0 ± 0.2** | 17.2 ± 0.9 | 20.2 ± 0.5 | 24.8 ± 0.5 | **31.7 ± 0.3** |
| | 10 | 38.6 ± 0.8 | 39.6 ± 0.4 | 56.0 ± 0.1 | **67.0 ± 0.4** | 36.3 ± 0.7 | 39.4 ± 0.9 | 47.3 ± 0.5 | **74.7 ± 0.1** | 33.8 ± 1.1 | 37.2 ± 0.6 | 45.1 ± 0.7 | **70.9 ± 1.3** |
| | 50 | 62.7 ± 0.5 | 57.7 ± 0.4 | 71.1 ± 0.2 | **77.3 ± 0.1** | 64.5 ± 0.6 | 62.8 ± 1.2 | 76.4 ± 0.4 | **89.2 ± 0.1** | 61.5 ± 0.6 | 61.6 ± 0.2 | 74.1 ± 0.6 | **88.5 ± 0.3** |
| CF-100 | 1 | 13.4 ± 0.3 | 12.9 ± 0.1 | 21.8 ± 0.4 | **32.5 ± 0.3** | 13.4 ± 0.5 | 11.5 ± 0.4 | 4.6 ± 0.1 | **31.7 ± 1.2** | 12.6 ± 0.6 | 10.1 ± 0.1 | 4.5 ± 0.2 | **27.8 ± 1.5** |
| | 10 | 38.7 ± 0.8 | 34.2 ± 0.3 | 47.0 ± 0.3 | **51.3 ± 0.1** | 47.0 ± 0.4 | 42.7 ± 0.5 | 53.4 ± 0.3 | **64.8 ± 0.0** | 47.5 ± 0.5 | 44.2 ± 0.5 | 54.0 ± 0.3 | **66.2 ± 0.0** |
| | 50 | 53.8 ± 0.4 | 52.2 ± 0.3 | 55.3 ± 0.2 | **56.3 ± 0.3** | 60.0 ± 0.1 | 57.4 ± 0.2 | 64.0 ± 0.0 | **68.8 ± 0.2** | 62.2 ± 0.3 | 60.6 ± 0.2 | 65.8 ± 0.3 | **69.6 ± 0.2** |
| T-IN | 1 | 8.6 ± 0.2 | 11.8 ± 0.4 | 17.0 ± 0.4 | **27.0 ± 0.3** | 9.0 ± 0.1 | 13.5 ± 0.2 | 15.4 ± 0.6 | **24.2 ± 1.4** | 8.2 ± 0.2 | 12.8 ± 0.2 | 14.8 ± 0.5 | **23.2 ± 1.1** |
| | 10 | 33.9 ± 0.2 | 34.5 ± 0.5 | 41.2 ± 0.5 | **42.9 ± 0.1** | 37.7 ± 0.3 | 43.6 ± 0.5 | 48.4 ± 0.3 | **55.8 ± 0.1** | 41.3 ± 0.1 | 47.0 ± 0.1 | 50.2 ± 0.1 | **58.1 ± 0.4** |
| | 50 | 46.6 ± 0.2 | 46.3 ± 0.2 | 47.1 ± 0.2 | **47.4 ± 0.2** | 52.2 ± 0.0 | 53.4 ± 0.3 | 57.4 ± 0.2 | **59.9 ± 0.0** | 55.9 ± 0.3 | 57.1 ± 0.1 | 59.2 ± 0.2 | **61.5 ± 0.0** |
| IN-1k | 1 | 3.4 ± 0.1 | 2.5 ± 0.0 | 5.2 ± 0.1 | **10.8 ± 0.0** | 2.1 ± 0.1 | 2.8 ± 0.2 | 5.9 ± 0.1 | **5.9 ± 0.2** | 1.6 ± 0.0 | 3.0 ± 0.4 | **6.8 ± 0.1** | 5.9 ± 0.2 |
| | 10 | 22.6 ± 0.7 | 12.7 ± 0.3 | 20.4 ± 0.3 | **32.2 ± 0.1** | 35.7 ± 0.2 | 31.1 ± 0.1 | 41.1 ± 0.2 | **48.5 ± 0.3** | 42.6 ± 0.4 | 38.3 ± 0.5 | 46.2 ± 0.3 | **48.5 ± 0.3** |
| | 50 | 37.8 ± 0.4 | 36.0 ± 0.3 | 38.8 ± 0.6 | **52.9 ± 0.4** | 51.6 ± 0.2 | 49.5 ± 0.2 | 55.3 ± 0.2 | **61.3 ± 0.1** | 60.3 ± 0.1 | 58.4 ± 0.1 | 62.5 ± 0.1 | **61.3 ± 0.1** |

# J  RELA Algorithm

---

**Algorithm 1** Adaptive Loss Weighting Algorithm for RELA and Self-Supervised Learning

---

**Require:** Number of training steps $T$, Initial $\ell_s = 2.0$, Initial $\ell_f = 1.0$, Initial $\lambda = 1$
1: **for** each training step $t = 1$ **to** $T$ **do**
2:   **if** $\lambda = 1$ **then**
3:     $\ell_c \leftarrow$ Value of $\mathcal{L}_{\text{RELA}}$ {Retrieve the current loss value from the optimization process}
4:     $\ell_f \leftarrow 0.999 \times \ell_f + 0.001 \times \ell_c$ {Calculate the short-term loss}
5:     $\ell_s \leftarrow 0.99 \times \ell_s + 0.01 \times \ell_f$ {Calculate the long-term loss}
6:   **end if**
7:   **if** $\exp(- \max\{\ell_s - \ell_f, 0\}) \geq 0.995$ **then**
8:     $\lambda \leftarrow 0$ {RELA learning is converged and over}
9:   **end if**
10:   $\mathcal{L} \leftarrow \lambda \cdot \mathcal{L}_{\text{RELA}} + (1 - \lambda) \cdot \mathcal{L}_{\text{SSL}}$ {Control the RELA and the SSL states using $\lambda$}
11: **end for**

---

In essence, this algorithm is designed to detect the convergence of RELA. Upon convergence, it transitions to the original algorithm for self-supervised learning. This implicitly segments the learning procedure into two phases: the fast stage (RELA) and the slow stage (SSL). Moreover, Figure 5 depicts the dynamics of the training process.

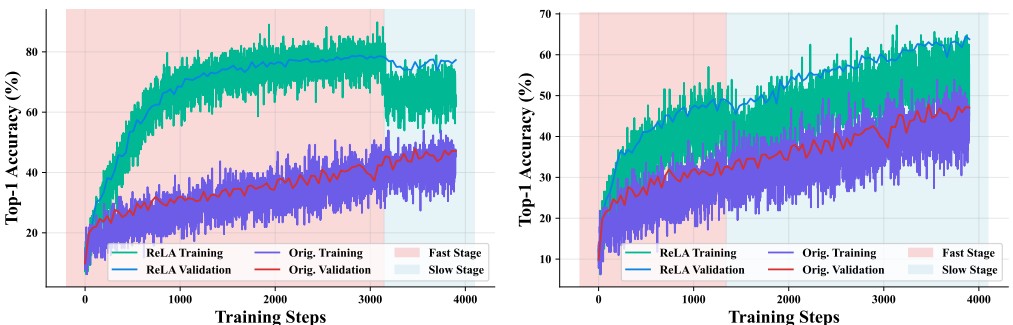

| (a) RELA with CF10-T as the prior model | (b) RELA with Rand. as the prior model |
|---|---|

Figure 5: **Comparison of training dynamics between RELA and the original (Orig.) BYOL algorithm**.

# K  Experimental Details

## K.1  Detailed Setup for Experiments in Section 3.2

We show the detailed setup for the experiments in Section 3.2 in Tables 6 and 7.

Table 6: **Architecture of the simple neural network model.**

| Layer | Type | Input Units | Output Units |
|---|---|---|---|
| Input Layer | - | 2 | - |
| Hidden Layer | Fully Connected (Linear) | 2 | 50 |
| | Activation (ReLU) | - | - |
| Output Layer | Fully Connected (Linear) | 50 | 1 |
| | Activation (Sigmoid) | - | - |
| Loss Function | Mean Squared Error (MSE) | - | - |

Table 7: **Optimization parameters for the simple neural network model.**

| Parameter | Value |
|---|---|
| Training Steps | 1000 |
| Batch Size | 1 |
| Optimizer | Stochastic Gradient Descent (SGD) |
| Learning Rate | 0.002 |
| Momentum | 0.98 |

## K.2 Detailed Setup for Experiments in Section 5

**Training details for self-supervised learning methods.** We show the details of training in Table 8.

Table 8: **Training parameters and optimizer settings for self-supervised learning methods.**

| Parameter | Value |
|---|---|
| Epochs | 100 |
| Optimizer | AdamW |
| Learning Rate | 0.001 |
| Weight Decay | 0.01 |

**Linear evaluation details.** We show the details of training the linear model in Table 9.

## L  Batch PCA Reduction

To begin, we provide a full and rigorous mathematical framework for Principal Component Analysis (PCA).

### L.1  Principal Component Analysis

Given a data matrix $\mathbf{Y} \in \mathbb{R}^{n \times d}$ and the desired number of components $k$, where $k \leq d$, PCA aims to extract a reduced data matrix $\mathbf{Y}_{\text{reduced}} \in \mathbb{R}^{n \times k}$ that retains the maximum variance from the original dataset.

First, the data needs to be centered by subtracting the mean of each feature. This can be mathematically represented as:

$$\mathbf{Y}_{\text{centered}} = \mathbf{Y} - \frac{1}{n}\mathbf{1}_n\mathbf{1}_n^T\mathbf{Y}$$

where $\mathbf{1}_n$ is a column vector of ones with length $n$. Next, the covariance matrix of the centered data is computed:

$$\mathbf{C} = \frac{1}{n-1}\mathbf{Y}_{\text{centered}}^T\mathbf{Y}_{\text{centered}}$$

Table 9: **Linear evaluation parameters for various datasets.**

| Dataset | Batch Size | Linear Model | Loss Function |
|---|---|---|---|
| CIFAR-10 | 128 | nn.Linear(*feature_dim*, 10) | CrossEntropyLoss() |
| CIFAR-100 | 128 | nn.Linear(*feature_dim*, 100) | CrossEntropyLoss() |
| Tiny-ImageNet | 128 | nn.Linear(*feature_dim*, 200) | CrossEntropyLoss() |
| ImageNet-1K | 1024 | nn.Linear(*feature_dim*, 1000) | CrossEntropyLoss() |
| Optimizer | | Adam (learning rate: 3e-4) | |

To identify the principal components, eigendecomposition is performed on the covariance matrix:

$$\mathbf{C} = \mathbf{V}\mathbf{\Lambda}\mathbf{V}^T$$

where $\mathbf{V}$ is the matrix of eigenvectors and $\mathbf{\Lambda}$ is a diagonal matrix of eigenvalues. The eigenvectors are then sorted in descending order based on their corresponding eigenvalues. The top $k$ eigenvectors are selected to form the projection matrix:

$$\mathbf{W} = [\mathbf{v}_1, \mathbf{v}_2, \ldots, \mathbf{v}_k]$$

where $\mathbf{v}_i$ represents the $i$-th eigenvector. Finally, the centered data is projected onto the new subspace:

$$\mathbf{Y}_{\text{reduced}} = \mathbf{Y}_{\text{centered}}\mathbf{W}$$

The resulting reduced data matrix $\mathbf{Y}_{\text{reduced}}$ contains the principal components of the original data. Each principal component is a linear combination of the original features, designed to capture as much variance as possible in the reduced space.

---

**Algorithm 2** Batch PCA Reduction

---

**Require:** Data matrix $\mathbf{Y} \in \mathbb{R}^{n \times d}$, Number of components $k$
**Ensure:** Reduced data matrix $\mathbf{Y}_{\text{reduced}} \in \mathbb{R}^{n \times k}$
 1: $n \leftarrow$ number of rows in $\mathbf{Y}$
 2: $m \leftarrow$ BATCHSIZE {Pre-set Batch size}
 3: $\mu \leftarrow \frac{1}{n}\sum_{i=1}^{n} \mathbf{Y}_{i,\cdot}$ {Compute the mean of $\mathbf{Y}$}
 4: $\mathbf{Y}_{\text{centered}} \leftarrow \mathbf{Y} - \mu$ {Center the data}
 5: $\mathbf{\Sigma} \leftarrow \mathbf{0}_{d \times d}$ {Initialize the covariance matrix}
 6: **for** $i = 0$ **to** $n$ **step** $m$ **do**
 7: $\quad j \leftarrow \min(i + m, n)$
 8: $\quad \mathbf{B} \leftarrow \mathbf{Y}_{\text{centered}}[i:j, \cdot]$
 9: $\quad \mathbf{\Sigma} \leftarrow \mathbf{\Sigma} + \mathbf{B}^{\top}\mathbf{B}$ {Update the covariance matrix}
10: **end for**
11: $\mathbf{\Sigma} \leftarrow \frac{\mathbf{\Sigma}}{n-1}$ {Normalize the covariance matrix}
12: $\mathbf{V} \leftarrow$ eigenvectors of $\mathbf{\Sigma}$ corresponding to the largest $k$ eigenvalues
13: $\mathbf{Y}_{\text{reduced}} \leftarrow \mathbf{0}_{n \times k}$ {Initialize the reduced data matrix}
14: **for** $i = 0$ **to** $n$ **step** $m$ **do**
15: $\quad j \leftarrow \min(i + m, n)$
16: $\quad \mathbf{B} \leftarrow \mathbf{Y}_{\text{centered}}[i:j, \cdot]$
17: $\quad \mathbf{Y}_{\text{reduced}}[i:j, \cdot] \leftarrow \mathbf{B}\mathbf{V}$
18: **end for**
19: **return** $\mathbf{Y}_{\text{reduced}}$

---

Then we prove that the batch PCA we utilized here is exactly same to the standard PCA.

*Proof.* To prove that the Batch PCA algorithm achieves the same result as the standard PCA, we need to show that the covariance matrix and the reduced data matrix computed by the Batch PCA are equivalent to those computed by the standard PCA.

First, let's show that the covariance matrix $\mathbf{\Sigma}$ computed in the Batch PCA is equivalent to the covariance matrix $\mathbf{C}$ in the standard PCA.

In the standard PCA, the covariance matrix is computed as:

$$\mathbf{C} = \frac{1}{n-1} \mathbf{Y}_{\text{centered}}^T \mathbf{Y}_{\text{centered}}$$

In the Batch PCA, the covariance matrix is computed as:

$$\begin{aligned}
\mathbf{\Sigma} &= \frac{1}{n-1} \sum_{i=0}^{n-1} \mathbf{B}_i^T \mathbf{B}_i \\
&= \frac{1}{n-1} \left( \mathbf{B}_0^T \mathbf{B}_0 + \mathbf{B}_1^T \mathbf{B}_1 + \ldots + \mathbf{B}_{b-1}^T \mathbf{B}_{b-1} \right) \\
&= \frac{1}{n-1} \mathbf{Y}_{\text{centered}}^T \mathbf{Y}_{\text{centered}}
\end{aligned}$$

where $\mathbf{B}_i$ is the $i$-th batch of the centered data matrix $\mathbf{Y}_{\text{centered}}$, and $b$ is the number of batches.

Therefore, the covariance matrix computed by the Batch PCA is equivalent to the covariance matrix computed by the standard PCA.

Next, let's show that the reduced data matrix $\mathbf{Y}_{\text{reduced}}$ computed in the Batch PCA is equivalent to the one computed in the standard PCA.

In the standard PCA, the reduced data matrix is computed as:

$$\mathbf{Y}_{\text{reduced}} = \mathbf{Y}_{\text{centered}} \mathbf{W}$$

where $\mathbf{W}$ is the matrix of the top $k$ eigenvectors of the covariance matrix.

In the Batch PCA, the reduced data matrix is computed as:

$$\begin{aligned}
\mathbf{Y}_{\text{reduced}} &= \begin{bmatrix} \mathbf{B}_0 \mathbf{V} \\ \mathbf{B}_1 \mathbf{V} \\ \vdots \\ \mathbf{B}_{b-1} \mathbf{V} \end{bmatrix} \\
&= \begin{bmatrix} \mathbf{Y}_{\text{centered}}[0:m, \cdot] \\ \mathbf{Y}_{\text{centered}}[m:2m, \cdot] \\ \vdots \\ \mathbf{Y}_{\text{centered}}[(b-1)m:n, \cdot] \end{bmatrix} \mathbf{V} \\
&= \mathbf{Y}_{\text{centered}} \mathbf{V}
\end{aligned}$$

where $\mathbf{V}$ is the matrix of the top $k$ eigenvectors of the covariance matrix $\mathbf{\Sigma}$, and $m$ is the batch size.

Since we have shown that the covariance matrices $\mathbf{C}$ and $\mathbf{\Sigma}$ are equivalent, their eigenvectors $\mathbf{W}$ and $\mathbf{V}$ are also equivalent. Therefore, the reduced data matrix computed by the Batch PCA is equivalent to the one computed by the standard PCA.

In conclusion, we have proven that the Batch PCA algorithm achieves the same result as the standard PCA by showing that the covariance matrix and the reduced data matrix computed by the Batch PCA are equivalent to those computed by the standard PCA. $\square$

## M   Analyze the Practical Data Augmentations

Here we presents a rigorous mathematical proof demonstrating that higher data augmentation intensity leads to increased overlap between samples from different classes. By modeling the sample distributions and augmentation process, we show that the variance of the augmented distributions increases with augmentation strength, resulting in wider and more overlapping distributions. We provide an exact calculation of the intersection point and approximate the overlap area using the cumulative distribution function of the standard normal distribution. Our theoretical analysis confirms the positive correlation between data augmentation intensity and inter-class overlap rate.

### M.1   Assumptions and Definitions

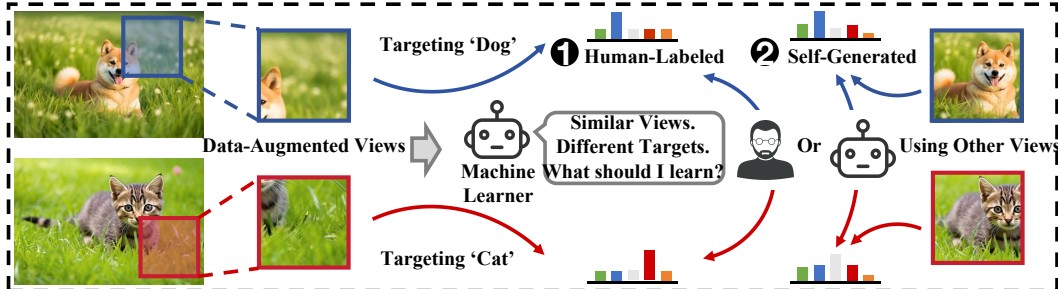

Figure 6: **A scenario of similar views targeting different targets**. In some cases, we may randomly crop two similar views from the original images as augmentations and input them into the machine learning model. This can lead to confusion in the model due to the differing targets assigned by humans or generated autonomously.

**Assumption 1** . *There are two classes $C_1$ and $C_2$, with samples drawn from probability distributions $P_1$ and $P_2$, respectively. The data augmentation intensity is denoted by $\alpha$.*

**Definition 10 (Data Distribution and Augmentation)** . *The samples are satisfied $X_1 \sim P_1$, $X_2 \sim P_2$. The augmented samples are represented as $X_1' = X_1 + \epsilon$ and $X_2' = X_2 + \epsilon$, where $\epsilon$ denotes the augmentation perturbation following the distribution $Q_\alpha$. The variance of $Q_\alpha$ is proportional to $\alpha$.*

## M.2 Augmented Distributions

Let $P_{1,\alpha}$ and $P_{2,\alpha}$ denote the augmented sample distributions:

$$P_{1,\alpha}(x') = (P_1 * Q_\alpha)(x') \tag{23}$$
$$P_{2,\alpha}(x') = (P_2 * Q_\alpha)(x') \tag{24}$$

where $*$ represents the convolution operation.

**Variance of Augmented Distributions**  Let $\sigma_\alpha^2$ be the variance of $Q_\alpha$, with $\sigma_\alpha^2 = k\alpha$, where $k$ is a constant. The variances of the augmented distributions are:

$$\sigma_{P_{1,\alpha}}^2 = \sigma_{P_1}^2 + \sigma_\alpha^2 = \sigma_{P_1}^2 + k\alpha \tag{25}$$
$$\sigma_{P_{2,\alpha}}^2 = \sigma_{P_2}^2 + \sigma_\alpha^2 = \sigma_{P_2}^2 + k\alpha \tag{26}$$

**Overlap Rate Definition**  Let $R$ denote the overlap region. The overlap rate $O(\alpha)$ is defined as:

$$O(\alpha) = \int_R P_{1,\alpha}(x')\,dx' + \int_R P_{2,\alpha}(x')\,dx' \tag{27}$$

## M.3 Increased Variance Leads to Increased Overlap

As $\alpha$ increases, $\sigma_\alpha^2$ increases, resulting in larger variances of $P_{1,\alpha}$ and $P_{2,\alpha}$. This makes the distributions wider and more dispersed, increasing the overlap region.

**Specific Derivation for One-Dimensional Gaussian Distributions**  Assume $P_1$ and $P_2$ are two Gaussian distributions:

$$P_1(x) = \mathcal{N}(\mu_1, \sigma_{P_1}^2) \tag{28}$$
$$P_2(x) = \mathcal{N}(\mu_2, \sigma_{P_2}^2) \tag{29}$$

The augmented distributions are:

$$P_{1,\alpha}(x') = \mathcal{N}(\mu_1, \sigma_{P_1}^2 + k\alpha) \tag{30}$$

$$P_{2,\alpha}(x') = \mathcal{N}(\mu_2, \sigma_{P_2}^2 + k\alpha) \tag{31}$$

**Exact Calculation of Intersection Point**   Equating the two Gaussian probability density functions and solving for $x$:

$$\frac{1}{\sqrt{2\pi(\sigma_{P_1}^2 + k\alpha)}} \exp\left(-\frac{(x - \mu_1)^2}{2(\sigma_{P_1}^2 + k\alpha)}\right) = \frac{1}{\sqrt{2\pi(\sigma_{P_2}^2 + k\alpha)}} \exp\left(-\frac{(x - \mu_2)^2}{2(\sigma_{P_2}^2 + k\alpha)}\right) \tag{32}$$

Simplifying the equation yields an analytical solution for the intersection point. To simplify the analysis, we assume $\sigma_{P_1}^2 = \sigma_{P_2}^2$, in which case the intersection point is $(\mu_1 + \mu_2)/2$.

**Area of Overlap Region**   Let $\Delta\mu = |\mu_1 - \mu_2|$. The overlap region can be represented using the cumulative distribution function (CDF) of the standard normal distribution, denoted as $\Phi(z)$:

$$\Phi(z) = \frac{1}{2}\left[1 + \mathrm{erf}\left(\frac{z}{\sqrt{2}}\right)\right] \tag{33}$$

The variance of the augmented distributions is $\sigma_\alpha^2 = \sigma_{P_1}^2 + k\alpha$. Therefore, the approximate area of the overlap region is:

$$O(\alpha) \approx 2\Phi\left(\frac{\Delta\mu}{\sqrt{2(\sigma_{P_1}^2 + k\alpha)}}\right) - 1 \tag{34}$$

## M.4   Conclusion

**Theorem 4 .** *As the data augmentation intensity $\alpha$ increases, the overlap rate $O(\alpha)$ between samples from different classes increases.*

*Proof.* Increasing $\alpha$ leads to an increase in the variance of the sample distributions, making them wider and more likely to overlap.

Specifically, increasing $\alpha$ increases $\sigma_{P_1}^2 + k\alpha$ in the denominator, thereby decreasing the argument of $\Phi(\cdot)$. Since $\Phi(z)$ increases as $z$ decreases for $z > 0$, $O(\alpha)$ increases as $\alpha$ increases.   □

In summary, we have rigorously proven the positive correlation between data augmentation intensity and inter-class overlap rate from a statistical and probabilistic perspective.

## M.5   Empirical Analysis for Real-world Datasets

Beyond the theoretical analysis for a simple case, we also provide empirical analysis for four real-world datasets under different intensities of data augmentation. We utilize TSNE [61] to visualize the (augmented) samples in Figure 7. All the results demonstrate that:

1. as minscale increases from 0.02 to 0.5, the data points gradually change from being tightly clustered to more dispersed. This indicates that higher data augmentation intensity expands the range of sample distribution;

2. when minscale=0.02, the data points of the two classes exhibit significant overlap, and the boundary becomes blurred. In contrast, larger minscale values such as 0.2 and 0.5 allow for better separation between classes;

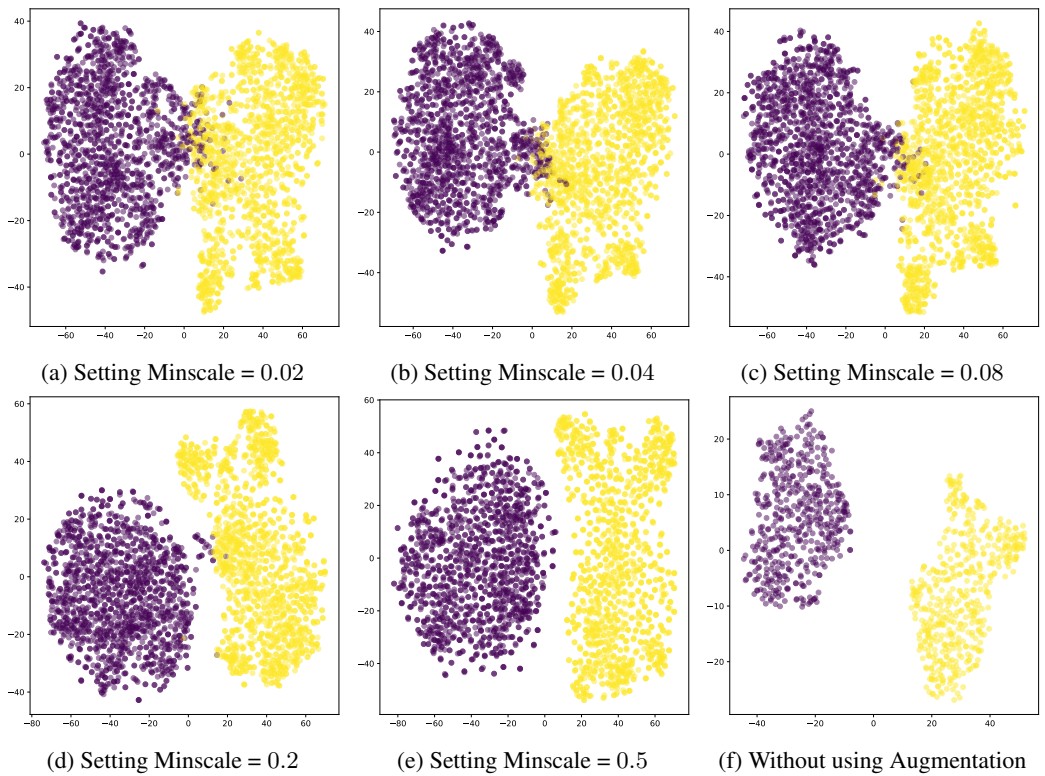

(a) Setting Minscale = 0.02     (b) Setting Minscale = 0.04     (c) Setting Minscale = 0.08

(d) Setting Minscale = 0.2     (e) Setting Minscale = 0.5     (f) Without using Augmentation

Figure 7: **Visualization of samples with varying levels of data augmentation intensity**. We visualize the samples of 2 classes from CIFAR-10 in 2-dimensional space, respectively setting minscale in `RandomResizeCrop` as $\{0.02, 0.04, 0.08, 0.2, 0.5\}$ and without data augmentation.

3. when data augmentation is not used (the last figure), there is a clear gap between the two classes, and the data points within each class are very compact. This suggests that the feature distribution of the original samples is relatively concentrated.

