# OpenReview forum: "Efficiency for Free: Ideal Data Are Transportable Representations"
_NeurIPS.cc/2024/Conference — NeurIPS 2024 poster_

### Official Review · Reviewer_7uZz · 2024-07-02

**Soundness:** 2
**Presentation:** 2
**Contribution:** 3
**Rating:** 6
**Confidence:** 3

**Summary:**

This paper considers data set distillation to a more concise form for purposes of representational efficiency, partly as an attempt to consider what forms of deployment might benefit from such forms and more importantly seeking to offer a universal form of translator to extract such compact representations from diverse data sets and their resulting influence towards training efficiency. It includes what appears to be an attempt at a nearly comprehensive survey of related literature which is itself a nice part of the paper.

**Strengths:**

- Originality
I didn't get a strong sense of significant originality, although that was sort of a vague sense and welcome further input from the authors. The five points were obviously clearly articulated in section one, I wondeer if you could color these with their specific relations to prior work?

    - Quality
The paper was not standout or anything, but for those with interest in this less common application it might have usefulness.

    - Clarity
The Table 1 was kind of awkwardly placed, it distracted from flow of paper and would possibly be better served as paragraphs instead of a table.

    - Significance
I had trouble interpreting the significance of the ReLA_D model for generic data set distillation, asking the authors for further input below. Otherwise a contribution of the work appeared to be associated with the demonstrations of impact of distillations towards training efficiency, which is sort of intuitive and probably more useful as a validation of their tool than any significant result.

**Weaknesses:**

The way the paper is written it almost appears that authors are approaching the concepts of dataset distillation considerations (eg in Table 1) as if such matters are unexplored in literature, which I would sort of find surprising. I think the paper would have been better structured as leading with the ReLA-D model convention for data set distillation as a more prominent part of the writeup, including more detail as to how that relates to prior work.

**Questions:**

I am partly fascinated by scope not addressed by the paper as to where these forms of distilled data sets might be of added benefit due to their less noisy representations.

More of a hypothetical question, no need to answer: Are there any known applications where data augmentation is known to interfere with implementations? Perhaps that might be a starting point of where valuable sources of impact of this form of distillation could be considered.

Can you talk further about the extent of novelty and related work for the ReLA-D tool?

**Limitations:**

I don't know if their is any strong interest in dataset distillations in practice where they may be used in production systems. If there are perhaps you could highlight furhter as it may make the importance of the paper more clear.

---

> ### Author Rebuttal · Authors · 2024-08-07
>
> > [W1] The way the paper is written it almost appears that authors are approaching the concepts of dataset distillation considerations (eg in Table 1) as if such matters are unexplored in literature, which I would sort of find surprising. I think the paper would have been better structured as leading with the ReLA-D model convention for data set distillation as a more prominent part of the writeup, including more detail as to how that relates to prior work.
> >
>
> [R1] Thank you for your valuable feedback. We will revise the manuscript to explicitly situate the ReLA-D model within the context of existing literature on dataset distillation. Additionally, we will enhance the discussion to clarify how our work builds upon and differentiates from prior research, as outlined in our [R3] below.
>
> > [Q1] I am partly fascinated by scope not addressed by the paper as to where these forms of distilled data sets might be of added benefit due to their less noisy representations.
> >
>
> [R2] Thank you for your feedback. We have conducted additional experiments and further explorations, as detailed in attached PDF and our response to Reviewer zbwg.
>
> In the attached PDF, we have included additional experiments covering: (a) the higher-resolution CelebA-HQ dataset (1024 × 1024); (b) a continual learning task; (c) comparisons with more state-of-the-art baselines; (d) reports on computation time and peak GPU memory usage; (e) a segmentation task on the VOC 2012 dataset; and (f) the larger ImageNet-21K dataset.
>
> > [Q2] More of a hypothetical question, no need to answer: Are there any known applications where data augmentation is known to interfere with implementations? Perhaps that might be a starting point of where valuable sources of impact of this form of distillation could be considered.
> >
>
> [R3] To the best of our knowledge, no such applications currently exist.
>
> > [Q3] Can you talk further about the extent of novelty and related work for the ReLA-D tool?
> >
>
> [R4] Additional discussions include the following:
>
> 1. As detailed in Appendix K, the SOTA dataset distillation methods generally demand a higher computational budget compared to training on the full dataset. For instance, to distill ImageNet-1K, the SOTA efficient methods [a,b,c] require a model that is well-trained on ImageNet-1K. In contrast, our ReLA can distill a dataset using less than the budget required for training a single epoch (refer to our analysis in Appendix K). The technical solution is detailed in Section 4.
> 2. Our ReLA is capable of distilling large datasets for both self-supervised and supervised learning tasks, whereas conventional methods are typically limited to distilling small datasets or are exclusively applicable to supervised learning.
>
> [a] Sun, Peng, et al. "On the diversity and realism of distilled dataset: An efficient dataset distillation paradigm." CVPR 2024.
>
> [b] Yin, Zeyuan, Eric Xing, and Zhiqiang Shen. "Squeeze, recover and relabel: Dataset condensation at imagenet scale from a new perspective." NeurlPS 2023 Spotlight.
>
> [c] Shao, Shitong, et al. "Generalized large-scale data condensation via various backbone and statistical matching." CVPR 2024 Highlight.

---

> > ### Comment · Reviewer_7uZz · 2024-08-08
> > **Update to score**
> >
> > Thank you for your responses author(s). Based on your comments, and particularly towards [R4], I have increased the contribution score from 2 to 3 and the overall score from 4 to 6.

---

> > > ### Comment · Reviewer_7uZz · 2024-08-08
> > >
> > > Perhaps the points addressed in your response [R4] should be more strongly articulated in the manuscript?

---

> > > > ### Author Response · Authors · 2024-08-10
> > > >
> > > > Thank you for your thoughtful feedback and for reconsidering the scores based on our responses. We will ensure that the points addressed are more clearly and prominently articulated in the revised manuscript to enhance the clarity and impact of our contribution. Your input is invaluable in helping us improve the quality of our work.

---

### Official Review · Reviewer_K3hg · 2024-07-08

**Soundness:** 2
**Presentation:** 2
**Contribution:** 2
**Rating:** 3
**Confidence:** 4

**Summary:**

This paper proposes to accelerate the training of self-supervised learning with a pre-trained teacher model and use the prediction of this teacher as a target. The method is motivated from a toy example that the training convergence is dominated by the variance of the random variable. Then, the authors conjure that maximizing the mutual information between samples and targets can accelerate the convergence, based on which the authors give their solution that adds a cosine similarity term between the outputs from the student model and the teacher to the base self-supervised learning loss. Experiments demonstrate that the proposed technique enhances the performance when there are only part of data.

**Strengths:**

1. The experiments are sufficient and satisfactory to demonstrate the effectiveness of the proposed method.
2. Most of the claims are backed up with theoretical analysis.

**Weaknesses:**

1. I do not think the writing style of the article is proper for a technical paper. Overall speaking, it is over packaged from the motivation throughout to the final solution. They try to introduce a big picture in the abstract and introduction parts. However, readers can hardly know what they actually do until reading the method part. I recommend introducing the technical solution first and then discussing the underlying motivations.
2. Technically, my major concern lies in the motivation of the proposed method. In Conjecture 1, the authors speculate that maximizing the mutual information between samples and targets can speed up training. This conjecture is based on an over simplified example that conducts classification for a Gaussian Mixture Model with 2 Gaussian distributions. It's true that diminishing the variance can result in faster training in this simple case, but it is far from the real-world cases, which would be much more complex. In this way, although the article involves extensive theoretical analysis, the core part, which connects the motivation and the final solution, is not well supported.
3. In fact, this conjecture, as well as the final solution that minimizes the cosine loss with sample predictions and targets generated by a well-trained teacher, is contradictory to many recent studies. For example, in RDED [a], hard patches are selected to form the synthetic samples. After all, hard patches should have larger classification loss, corresponding to samples closer to the decision boundary in the toy case. Recent studies like [b] and [c] also demonstrate that weak trajectory or teacher models may benefit learning more. Given these studies, the conjecture of this article seems not so plausible without substantial evidences.
4. Based on above analysis, I would like to believe the acceleration comes from additional supervision signals, which transforms a self-supervised learning problem to an almost fully supervised one, since a well-trained teacher is involved. If that is the case, the conclusion and technique proposed in the article would appear trivial, since similar conclusion and technique have been well studied even several years ago [d,e,f,g].
5. The authors analyze *the optimal properties of distilled data*. Here an assumption that specifies the distillation method is missing. After all, different methods would result in different distill data. Not all of them satisfy the proposed properties.
6. I do not think the introduction of dynamic dataset distillation is necessary. The authors randomly sample a portion of real data to form the training dataset and use different data in each epoch, which can be viewed as mini-batch training on a subset and is not closely related to dataset distillation. In fact, I think the settings in DiM [h] and [i] are more adhere to this concept. In other words, the setting in this article is actually not coupled with dataset distillation. Only using a subset of data can achieve satisfactory performance is the result of introduction of pre-trained teacher. The method is not well aligned with dataset distillation serving as the motivation. In this case, the results may not appear surprising since the method transforms a self-supervised learning problem to an almost fully supervised one.

[a] On the Diversity and Realism of Distilled Dataset: An Efficient Dataset Distillation Paradigm (Peng Sun et al., CVPR 2024)

[b] Dataset Quantization (Daquan Zhou & Kai Wang & Jianyang Gu et al., ICCV 2023)

[c] SelMatch: Effectively Scaling Up Dataset Distillation via Selection-Based Initialization and Partial Updates by Trajectory Matching (Yongmin Lee et al., ICML 2024)

[d] Self-Supervised Dataset Distillation for Transfer Learning (Dong Bok Lee & Seanie Lee et al., ICLR 2024)

[e] Boosting Self-Supervised Learning via Knowledge Transfer (Noroozi et al., CVPR 2018)

[f] Knowledge Distillation Meets Self-Supervision (Xu et al., ECCV 2020)

[g] SEED: Self-supervised Distillation For Visual Representation (Fang et al., ICLR 2021)

[h] DiM: Distilling Dataset into Generative Model (Kai Wang & Jianyang Gu et al., 2023)

[I] Data Distillation Can Be Like Vodka: Distilling More Times For Better Quality (Xuxi Chen & Yu Yang et al., ICLR 2024)

**Questions:**

Please refer to the weakness part above.

**Limitations:**

The authors have mentioned that Conjecture 1 remains speculative. However, I do think this is an important limitation that serves as the core of the whole method and article.

---

> ### Author Rebuttal · Authors · 2024-08-07
>
> > [W1] […] I recommend introducing the technical solution first and then discussing the underlying motivations.
> >
>
> [R1] Thank you for your feedback. We will revise the paper to present the technical solution earlier, enhancing clarity and focus.
>
> > [W2] Technically, my major concern lies in the motivation of the proposed method. In Conjecture 1, […] This conjecture is based on an over simplified example that conducts classification for a Gaussian Mixture Model with 2 Gaussian distributions. […]
> >
>
> [R2] We would like to clarify the following points:
>
> 1. Prior research efforts [n,o], including the Outstanding Paper at NeurIPS 2022, have utilized simplified mixtures of Gaussian distributions to study data efficiency and other properties, acknowledging the complexity of analyzing real-world data distributions. These studies typically extend theoretical insights to more complex datasets through empirical validation.
> 2. Our work provides theoretical insights into data properties relevant to efficient training. These insights do not suggest a reduction in the variance of real-world data but rather support our Conjecture 1, which forms the basis of our proposed technical solution. The validity of our solution is demonstrated in Section 5, where we present extensive experimental results on real-world datasets.
>
> > [W3] In fact, this conjecture, as well as the final solution that minimizes the cosine loss with sample predictions and targets generated by a well-trained teacher, is contradictory to many recent studies. For example, in RDED [a], hard patches are selected to form the synthetic samples. After all, hard patches should have larger classification loss, corresponding to samples closer to the decision boundary in the toy case. Recent studies like [b] and [c] also demonstrate that weak trajectory or teacher models may benefit learning more. […]
> >
>
> [R3] We would like to clarify the following points:
>
> 1. In fact, RDED [a] promotes the selection of easy patches (those with smaller classification loss, as indicated in their Equation (8)) and utilizes well-trained teacher models for target generation, as specified in their Algorithm 1. This methodology is consistent with our theoretical and empirical analyses and aligns with the recommendations found in several recent studies [e,f,g,j,k,l].
> 2. The papers reviewer referenced [b, c] utilize weak teachers for trajectory matching or patch selection in image generation. However, none of these studies advocate for the use of weak teachers in target generation.
>
> > [W4] Based on above analysis, I would like to believe the acceleration comes from additional supervision signals, which transforms a self-supervised learning problem to an almost fully supervised one, since a well-trained teacher is involved. […]
> >
>
> [R4] We would like to clarify the following points:
>
> 1. Unlike prior studies [a,b,e,f,g,j,k,l] that directly utilize a well-trained model specific to the dataset, our approach, ReLA, leverages any publicly available model to generate supervision signals. For instance, ReLA can use a model pre-trained on CIFAR-10 to generate supervision signals that accelerate training for BYOL on a 10% subset of ImageNet-1K, resulting in a 7.7% increase in accuracy compared to the original BYOL (see Sections 4 and 5).
> 2. Our aim is to propose a simple, effective, plug-and-play, and theoretically supported method to accelerate representation learning. In addition to empirical validation in our paper and similar evidence in prior studies [d,e,f,g,j,k,l], we also provide theoretical insights into how these generated supervision signals enhance the convergence rate of learning (see Section 3.2).
> 3. We have not transformed a self-supervised learning problem into an almost fully supervised one. The additional supervision loss is integrated with the original loss term and modulated by a dynamic coefficient (see Section 4.2). Furthermore, our ablation study in Section 5.2 illustrates that using only the supervision loss can result in catastrophic performance.
> 4. Moreover, our method can also be used to accelerate supervised learning, as demonstrated in our experiments and analysis in Appendix L.
>
> > [W5] The authors analyze *the optimal properties of distilled data*. […], different methods would result in different distill data. Not all of them satisfy the proposed properties.
> >
>
> [R5] We would like to clarify the following points:
>
> 1. Our analysis try to identify the ideal properties of efficient data to offer insights that support our proposed framework. Most existing dataset distillation methods fail to fully satisfy these properties.  The relationship between our analyzed properties and previous studies is discussed in Section 3.2 and Appendix H.
> 2. However, our theoretical results are consistent with the empirical results in many previous studies [e,f,g,j,k,l,m], which indicate that: (i) selecting/synthesizing samples that are easy or close to the class data mean, and (ii) using well-trained models to relabel samples, can achieve superior performance with fewer training steps.
>
> > [W6] I do not think the introduction of dynamic dataset distillation is necessary. […], the results may not appear surprising since the method transforms a self-supervised learning problem to an almost fully supervised one.
> >
>
> [R6] We would like to clarify the following points:
>
> 1. Our framework ReLA is defined under the concept of dynamic dataset distillation because ReLA involves dynamically editing samples and labels during training.
> 2. Our approach does not rely solely on the introduction of a pre-trained teacher, as detailed in our analysis in [R4].  Furthermore, our experiments in Section 5 show that introducing weak teachers also contributes positively to our framework.
> 3. In addition to self-supervised learning tasks, we demonstrate that our ReLA framework outperforms state-of-the-art dataset distillation methods in conventional supervised learning tasks (see Appendix L).

---

> > ### Comment · Reviewer_K3hg · 2024-08-09
> >
> > I would like to thank the authors for the detailed response. However, my main concerns are still not addressed. I understand that some papers have conducted theoretical analysis under the setting of GMM. There is still unaddressed controversy with some works. And, the technical contribution of the article still appears trivial to me.
> >
> > 1. In R3, the authors have clarified the points in RDED, DQ, and SelMatch. However, there are indeed some works advocating for the use of weak teachers in target generation, such as Fig. 2 in [Qin et al. 2024]. I perfectly understand that it showed on arXiv after the NeurIPS submission deadline. Nevertheless, their conclusions seem more convincing because it is tested on real-world cases instead of merely analyzed through simple GMMs. Especially in the case when there are only a small number of training samples, which is the main focus of this article for efficient training, weak teachers can be more useful. Taking this into consideration, I do think there exists a significant gap between the theoretical analysis and the practical cases.
> > 2. I also perfectly understand that merely using supervised loss is insufficient. However, the experimental enhancement is not surprising to me at all. Compared with the baseline, the method in this article introduces a pre-trained teacher model on large-scale data, whose representation is definitely useful especially in self-supervised problems. The thing this article actually does is to introduce a knowledge-distillation-like loss term to enhance the performance, which is a popular technique in practice.
> >
> > Taking these factors into consideration, I tend to maintain my original score.
> >
> > A Label is Worth a Thousand Images in Dataset Distillation, Qin et al., 2024

---

> > > ### Author Response · Authors · 2024-08-10
> > >
> > > > [Q1] In R3, the authors have clarified the points in RDED, DQ, and SelMatch. However, there are indeed some works advocating for the use of weak teachers in target generation, such as Fig. 2 in [Qin et al. 2024]. I perfectly understand that it showed on arXiv after the NeurIPS submission deadline. Nevertheless, their conclusions seem more convincing because it is tested on real-world cases instead of merely analyzed through simple GMMs. Especially in the case when there are only a small number of training samples, which is the main focus of this article for efficient training, weak teachers can be more useful. Taking this into consideration, I do think there exists a significant gap between the theoretical analysis and the practical cases.
> > > >
> > >
> > > [R1] We respectfully disagree with the reviewer and would like to clarify that there is no significant gap between the theoretical analysis and the empirical cases in our paper:
> > >
> > > 1. We drew insights from the theoretical analysis, then designed an effective and efficient technical ReLA framework, and empirically validated our proposed framework ReLA on extensive real-world datasets and downstream tasks.
> > >     1. Our insights align with the conclusions presented in Sections 4 and 6 of the referenced paper [1], which are derived from their experimental analysis. Specifically, we establish the insights articulated in Conjecture 1 and Definition 5 through our Theorems 1, 2, and 4. Furthermore, our insights emphasize the importance of employing labelers to provide ***informative*** targets for efficient learning, which corroborates the conclusion of paper [1], asserting the significance of incorporating “structured information.”
> > >     2. Moreover, our Theorem 4 indicates that using data consisting of both randomly selected samples and high-quality targets can lead to superior performance. This insight aligns with the empirical results presented in Table 1 of paper [1], where it is demonstrated that randomly selected samples, when labeled by a well-trained teacher, surpass the state-of-the-art SRe$^2$L method [2]. However, paper [1] does not provide a theoretical explanation for this observation.
> > >     3. The experimental results presented in Figure 2 of paper [1] do not sufficiently illustrate the inapplicability of our theoretical findings and insights. Although the paper [1] investigates the teacher model's role in generating distilled soft labels, as depicted in Figure 2, it fails to provide essential details regarding the methodology for producing the corresponding distilled samples or images. The omission of critical experimental parameters—such as the optimizer, learning rate, and the number of training epochs for the student model—compromises the ability to accurately compare their experimental setup with the conditions assumed in our theoretical analysis.
> > >     4. Moreover, the study presented in paper [1] lacks additional experiments to validate the phenomena observed in Figure 2, such as evaluations with alternative optimization algorithms or different distillation methods. Their current analysis is limited to a single illustrative case. It is also crucial to note that this paper [1] is a preprint submitted to ArXiv after the NeurIPS deadline, and its experimental details and methodologies have not yet undergone peer review.
> > > 2. The significant empirical gains we provided align well with our theoretical findings and insights. In fact, the primary objective of our theoretical analysis is to try to identify the ideal properties of data-efficient training in order to offer insights that support our Conjecture 1, Definition 5, and thereby our proposed technical framework ReLA. We included extensive empirical results (in Section 5) across various datasets, neural architectures, and learning algorithms, to support our findings: either a weak or strong prior model can create efficient data to significantly accelerate the training.
> > >     1. Small-scale CIFAR models improves ImageNet: ReLA can use a model pre-trained on CIFAR-10 to generate efficient data that accelerate training for BYOL on a 10% subset of ImageNet-1K, resulting in a 7.7% increase in accuracy compared to the original BYOL (see Table 2).
> > >     2. Using CLIP as a prior model, ReLA-aided BYOL enables training a ResNet-50 from scratch on 50% of the ImageNet-1K dataset, achieving performance that exceeds models trained on the full dataset.
> > >
> > > To the best of our knowledge, the presented empirical results and the acceleration provided to the community are novel and significant.

---

> > > ### Author Response · Authors · 2024-08-10
> > >
> > > > [Q2] I also perfectly understand that merely using supervised loss is insufficient. However, the experimental enhancement is not surprising to me at all. Compared with the baseline, the method in this article introduces a pre-trained teacher model on large-scale data, whose representation is definitely useful especially in self-supervised problems. The thing this article actually does is to introduce a knowledge-distillation-like loss term to enhance the performance, which is a popular technique in practice.
> > > >
> > >
> > > [R2] We would like to clarify the following points:
> > >
> > > 1. Our method does not depend on a teacher model pre-trained on large-scale data. As demonstrated in Table 2, even a weak prior model can generate data that significantly enhance training on large datasets, such as ImageNet-1K. Specifically,
> > >     1. ReLA illustrates the ability to leverage a model pre-trained on CIFAR-10—a relatively small dataset—to produce data that accelerates BYOL training on a 10% subset of ImageNet-1K, yielding a 7.7% improvement in accuracy over the original BYOL.
> > >     2. Even a randomly initialized and untrained model can provide sufficient acceleration. For example, when employing a randomly initialized model as the prior model in our ReLA framework to generate efficient data, it enhances BYOL training on a 10% subset of ImageNet-1K, resulting in a 6.1% increase in accuracy compared to the original BYOL. We believe this finding presents a novel contribution to the field.
> > > 2. Our proposed method ReLA is fundamentally different from knowledge distillation:
> > >     1. Knowledge distillation [3] typically employs a well-trained large model as the teacher, with the aim to transfer its knowledge to a smaller model to retain similar performance on the corresponding dataset. In contrast, our proposed method, ReLA, is a simple, effective, plug-and-play approach that leverages publicly available models on the internet as prior models to generate efficient data, which, combined with the ReLA loss, accelerates downstream learning tasks and achieves much better performance (see Sections 4.1 and 4.2).
> > >         - ReLA imposes no restrictions on the architecture, scale, or pre-trained datasets of the prior models used for generating efficient data. To validate the feasibility of this approach, we tested the effectiveness of various prior models with different representational capacities, knowledge, and architectures, as shown in Tables 1 and 2.
> > >         - To respond to the reviewers' feedback, we conducted additional evaluations on various downstream tasks (refer to the supplementary PDF).
> > >     2. In ReLA, the dynamic coefficient used in the loss function during downstream task model training is critical. This coefficient dynamically decays over the course of training, particularly for the ReLA loss term related to efficient data. The rationale behind this decay is that the primary function of the efficient data is to provide 'rapid guidance' during the early stages of model training (see Section 4.2). Our ablation study, shown in Figure 4(b), further demonstrates the necessity of this dynamic decay, especially when the prior model used to generate the efficient data is weak. This approach is fundamentally different from the primary objective of knowledge distillation, which focuses on transferring the teacher model’s capabilities to the student model.
> > >     3. In the knowledge distillation process [3], each augmented view of the data typically necessitates corresponding soft labels from the teacher model. This enables the student model to emulate the teacher's behavior across various transformations of the input data. However, this method results in a computational cost that increases linearly with the number of training epochs. Specifically, if the number of training epochs is \(N\), the computational overhead for the teacher model scales to \(N\) times the cost of performing inference on the entire dataset. In contrast, as detailed in Appendix K, ReLA requires only a single inference pass of the prior model over the entire dataset to generate efficient data. Further supporting evidence is provided in the attached PDF.
> > >
> > > [1] Qin, Tian, Zhiwei Deng, and David Alvarez-Melis. "A Label is Worth a Thousand Images in Dataset Distillation." 2024.
> > >
> > > [2] Yin, Zeyuan, Eric Xing, and Zhiqiang Shen. "Squeeze, recover and relabel: Dataset condensation at imagenet scale from a new perspective." NeurlPS 2023 Spotlight.
> > >
> > > [3] Hinton, Geoffrey, Oriol Vinyals, and Jeff Dean. "Distilling the knowledge in a neural network." *2015*.

---

> > > ### Author Response · Authors · 2024-08-12
> > >
> > > We express our sincere appreciation for your insightful feedback and thorough review of our paper. Your comments have been instrumental in enhancing our work and refining the proposed ReLA method. In response to the specific questions ([Q1] and [Q2]) you raised, we have provided detailed explanations to address each concern comprehensively. Below, we summarize our key responses:
> > >
> > > - **[R1] We drew insights from the theoretical analysis, designed an effective and efficient ReLA framework, and empirically validated it on extensive real-world datasets and downstream tasks. Our theoretical analysis and observed empirical results are well-aligned, motivating our ReLA design, the key contribution of the manuscript:**
> > >     - Our theoretical insights, specifically detailed in Conjecture 1 and Definition 5, which indicate the importance of ***informative*** targets, are consistent with the conclusions drawn from the experimental analyses in Sections 4 and 6 of the referenced paper [1].
> > >     - Furthermore, Theorem 4 suggests that using a combination of randomly selected samples and high-quality targets enhances performance, a finding corroborated by the empirical results in Table 1 of paper [1].
> > >     - The experimental results shown in Figure 2 of paper [1] lack crucial experimental parameters and an introduction of the image distillation method. Hence, it is difficult to dismiss our theoretical results based solely on the empirical outcomes in Figure 2. Additionally, their analysis is confined to a single illustrative case, without further experiments to validate the observed phenomena.
> > >     - Our substantial empirical gains align well with our theoretical predictions and insights. We provided extensive empirical results (Section 5) across various datasets, neural architectures, and learning algorithms, demonstrating that both weak and strong prior models can generate efficient data, significantly accelerating training through our ReLA.
> > >     - Note that the empirical understanding preprint [1] was appeared after the NeurIPS 2024 submission deadline. Though we may share some overlapping observations (namely our theoretical analysis and [1]’s empirical study), such an overlap only strengthens our core ReLA contribution. We will incorporate discussions related to the findings presented in paper [1] into our manuscript.
> > > - **[R2] Our method is independent of a pre-trained teacher model on large-scale data and fundamentally differs from knowledge distillation:**
> > >     - Knowledge distillation [2] typically uses a well-trained, large model as the teacher to transfer knowledge to a smaller model, aiming to achieve comparable performance on the same dataset.
> > >     - In contrast, our ReLA approach offers 'rapid guidance' by leveraging efficient data during the initial stages of model training (see Section 4.2). ReLA allows various prior models, including weak models or even randomly initialized and untrained models, to generate efficient data that significantly enhance training on large datasets like ImageNet-1K.
> > >
> > > In response to the positive feedback from reviewers 7uZz and zbwg, who have acknowledged the merits of ReLA and subsequently increased their scores, we respectfully seek your final assessment. If our rebuttal has adequately addressed your concerns and highlighted the significance of our proposed method, we kindly request that you consider revising your score accordingly. An increased score is critically important to our work at this stage.
> > >
> > > We remain committed to addressing any remaining concerns and are open to further discussions or clarifications. Your valuable feedback has been instrumental in refining our research, and we appreciate the opportunity to enhance our work based on your input. Thank you for your time and efforts throughout the review process. We look forward to your further feedback.
> > >
> > > [1] Qin, Tian, Zhiwei Deng, and David Alvarez-Melis. "A Label is Worth a Thousand Images in Dataset Distillation." 2024.
> > >
> > > [2] Hinton, Geoffrey, Oriol Vinyals, and Jeff Dean. "Distilling the knowledge in a neural network." *2015*.

---

> > > > ### Comment · Reviewer_K3hg · 2024-08-14
> > > >
> > > > Thanks for the detailed response and I really appreciate it. However, I still think the increasing of accuracy largely comes from the extra guidance of the teacher model. No matter whether it is trained on a large dataset, it is a well pre-trained teacher and bringing it to unsupervised tasks would be helpful. Thus, it is hard to say that the increasing of accuracy is due to the theorems in this article.
> > > >
> > > > In other words, after reading the article, I cannot gain additional insights. After all, if we want to boost an unsupervised training task, bringing a trained teacher model is an easy and straightforward solution for us to try. I understand that the operations in this article may not be exactly the same as previous ones, but I do not think there is substantial difference in the overall framework.
> > > >
> > > > Therefore, I still choose to maintain my score. I understand that there may be some divergence with other reviewers. I would respect the final decision made by the area chairs.

---

> ### Author Response · Authors · 2024-08-07
> **Reference**
>
> [a] Sun, Peng, et al. "On the diversity and realism of distilled dataset: An efficient dataset distillation paradigm." CVPR 2024.
>
> [b] Zhou, Daquan, et al. "Dataset quantization." ICCV 2023.
>
> [c] Lee, Yongmin, and Hye Won Chung. "SelMatch: Effectively scaling up dataset distillation via selection-based initialization and partial updates by trajectory matching." ICML 2024.
>
> [d] Lee, Dong Bok, et al. "Self-supervised dataset distillation for transfer learning." ICLR 2024.
>
> [e] Noroozi, Mehdi, et al. "Boosting self-supervised learning via knowledge transfer." CVPR 2018.
>
> [f] Xu, Guodong, et al. "Knowledge distillation meets self-supervision.” ECCV 2020.
>
> [g] Fang, Zhiyuan, et al. "Seed: Self-supervised distillation for visual representation.” ICLR 2021.
>
> [h] Wang, Kai, et al. "Dim: Distilling dataset into generative model." Preprint 2023.
>
> [i] Chen, Xuxi, et al. "Data distillation can be like vodka: Distilling more times for better quality." ICLR 2024
>
> [j] Yin, Zeyuan, Eric Xing, and Zhiqiang Shen. "Squeeze, recover and relabel: Dataset condensation at imagenet scale from a new perspective." NeurlPS 2023 Spotlight.
>
> [k] Shao, Shitong, et al. "Generalized large-scale data condensation via various backbone and statistical matching." CVPR 2024 Highlight.
>
> [l] Shao, Shitong, et al. "Elucidating the Design Space of Dataset Condensation.” Preprint 2024.
>
> [m] Zhao, Bo, and Hakan Bilen. "Dataset condensation with distribution matching." WACV 2023.
>
> [n] Sorscher, Ben, et al. "Beyond neural scaling laws: beating power law scaling via data pruning." NeurlPS 2022 Outstanding Paper.
>
> [o] Loureiro, Bruno, et al. "Learning gaussian mixtures with generalized linear models: Precise asymptotics in high-dimensions.” NeurlPS 2021.

---

### Official Review · Reviewer_zbwg · 2024-07-09

**Soundness:** 4
**Presentation:** 3
**Contribution:** 4
**Rating:** 7
**Confidence:** 4

**Summary:**

The paper addresses the scalability constraints in representation learning by proposing a novel Representation Learning Accelerator (RELA). Current paradigms, focusing separately on self-supervised learning and dataset distillation, overlook the potential of intermediate acceleration. The authors define ideal data properties for optimization and generalization, enabling effective transport of model-generated representations. RELA leverages a task- and architecture-agnostic public model to form a dynamic data subset, enhancing (self-)supervised learning. Empirical results show that using CLIP ViT B/16 as a prior model, RELA-aided BYOL can train a ResNet-50 from scratch with 50% of ImageNet-1K, surpassing full dataset performance. This approach improves representation learning efficiency, offering impactful implications for reducing data requirements and computational resources in model training.

**Strengths:**

[S1] The paper tackles critical issues in efficient training and dataset distillation (DD).

[S2] RELA introduces a novel approach in dataset distillation, effectively bridging representation learning with data-efficient methods.

[S3] The paper is clearly and effectively written.

[S4] Comprehensive theoretical analysis supports the proposed claims. The paper explores analytical concepts in dataset distillation, employing information theory principles.

[S5] Extensive experimental and ablation studies are conducted in both the main paper and appendix.

**Weaknesses:**

[W1] The paper lacks generalization ability to high-resolution datasets, such as 1Kx1K, which are common in practical datasets like clinical and aerial images.

[W2] The practical applications of dataset distillation are not discussed. Demonstrating applicability in neural architecture search (NAS), continual learning, federated learning, and privacy preservation would be valuable for both the community and real-world scenarios.

[W3] Comparisons with some state-of-the-art methods, including DREAM [a], DataDAM [b], and SeqMatch [c], are missing from Table 6. Including these comparisons would strengthen the evaluation of the proposed method.

[W4] An analysis of the computational costs is absent. It is crucial to examine the training costs, including both memory and training time, for the proposed process and state-of-the-art methods to ensure data-efficient training algorithms.

[W5] The theoretical analysis does not generalize to complicated data distributions, such as two arbitrary mixtures of Gaussian distributions or a Mixture of Generalized Gaussian distributions (MGG).

Upon addressing these weaknesses, I will consider changing my initial rating to strong accept.

-------------

References:

[a] Liu, Yanqing, et al. "Dream: Efficient dataset distillation by representative matching." Proceedings of the IEEE/CVF International Conference on Computer Vision. 2023.

[b] Sajedi, Ahmad, et al. "Datadam: Efficient dataset distillation with attention matching." Proceedings of the IEEE/CVF International Conference on Computer Vision. 2023.

[c] Du, Jiawei,  et al. "Sequential subset matching for dataset distillation." Advances in Neural Information Processing Systems 36 (2024).

**Questions:**

1. Can RELA be extended to ImageNet-21K? If so, please provide some experimental results.

2. How does the proposed framework perform on other downstream tasks, such as object detection or segmentation?

**Limitations:**

The limitations are briefly discussed in the appendix. For more limitations, refer to the Weaknesses and Questions sections.

---

> ### Author Rebuttal · Authors · 2024-08-07
>
> > [W1] The paper lacks generalization ability to high-resolution datasets, such as 1Kx1K, which are common in practical datasets like clinical and aerial images.
> >
>
> [R1] Thank you for your feedback. We have conducted additional experiments using the CelebA-HQ dataset (1024 $\times$ 1024). The results, presented in Table 1 of the attached PDF, indicate that our ReLA effectively accelerates self-supervised learning on high-resolution datasets.
>
> > [W2] The practical applications of dataset distillation are not discussed. Demonstrating applicability in neural architecture search (NAS), continual learning, federated learning, and privacy preservation would be valuable for both the community and real-world scenarios.
> >
>
> [R2] We would like to emphasize that the primary application of our ReLA method is to accelerate representation learning. This extends the scope of conventional dataset distillation, as traditional methods require more computational resources in distillation process than training on the full dataset, making them unsuitable for accelerating learning processes (refer to our detailed analysis in [R4]).
>
> Furthermore, we have applied ReLA to continual learning, with the experimental results presented in Table 2 of the attached PDF. These results demonstrate that ReLA outperforms the baseline in this context.
>
> > [W3] Comparisons with some state-of-the-art methods, including DREAM [a], DataDAM [b], and SeqMatch [c], are missing from Table 6. Including these comparisons would strengthen the evaluation of the proposed method.
> >
>
> [R3] We excluded comparisons with methods [a,b,c] because they are inefficient when dealing with large-scale datasets, such as ImageNet-1K, particularly with large backbone architectures like ResNet-18, as demonstrated in previous studies [d,e,f].
> Hence, we evaluate these methods [a,b,c] against our proposed ReLA on four datasets with smaller backbone networks. The results of these comparisons are presented in Table 3 of the attached PDF, demonstrating that ReLA consistently outperforms these methods.
>
> We will add these baselines into our manuscript.
>
> > [W4] An analysis of the computational costs is absent. It is crucial to examine the training costs, including both memory and training time, for the proposed process and state-of-the-art methods to ensure data-efficient training algorithms.
> >
>
> [R4] As detailed in Appendix K, the SOTA dataset distillation methods generally incur higher computational costs than training on the entire dataset. For example, the SOTA efficient method RDED [d] necessitates a fully trained model on ImageNet-1K for distillation. Additionally, several other SOTA methods [a,b,c] also demand higher computational budgets compared to training on the full dataset, as documented in their respective studies.
>
> In contrast, our proposed ReLA method can distill a dataset using less than the computational budget required for a single epoch of full dataset training (see Appendix K for a detailed analysis).
>
> The computational costs of ReLA, along with comparisons to baseline methods, are presented in Table 4 of the attached PDF. The experimental results indicate that while ReLA slightly increases computational time and peak GPU memory usage when using a partial dataset for training, it significantly enhances accuracy, surpassing the performance achieved by training on the entire dataset.
>
> We will add these comparisons into our manuscript.
>
> > [W5] The theoretical analysis does not generalize to complicated data distributions, such as two arbitrary mixtures of Gaussian distributions or a Mixture of Generalized Gaussian distributions (MGG).
> >
>
> [R5] The theoretical analysis presented in this paper can indeed be generalized to arbitrary mixtures of Gaussian distributions. Although Section 3 exemplifies this with a specific mixture of Gaussian distributions, the proofs and theoretical framework detailed in Appendices B and C provide a generalized approach applicable to any mixture of two Gaussian distributions, regardless of their means and variances.
>
> > [Q1] Can RELA be extended to ImageNet-21K? If so, please provide some experimental results.
> >
>
> [R6] The ImageNet-1K dataset used in this study is already a substantial and challenging benchmark in the field of dataset distillation [a,b,c,d,e,f]. We extended our experiments to the ImageNet-21K dataset. As shown in Table 6 in the attached PDF, our ReLA approach also accelerates training for BYOL on ImageNet-21K, achieving higher performance with the same data usage as the original BYOL.
>
> > [Q2] How does the proposed framework perform on other downstream tasks, such as object detection or segmentation?
> >
>
> [R7] We evaluate the pre-trained models on a downstream segmentation task, as detailed in Table 5 of the attached PDF. The results demonstrate that models trained with our ReLA framework outperform baseline methods in both classification and segmentation tasks.
>
> [a] Liu, Yanqing, et al. "Dream: Efficient dataset distillation by representative matching." ICCV 2023.
>
> [b] Sajedi, Ahmad, et al. "Datadam: Efficient dataset distillation with attention matching." ICCV 2023.
>
> [c] Du, Jiawei, Qin Shi, and Joey Tianyi Zhou. "Sequential subset matching for dataset distillation." NeurIPS 2023.
>
> [d] Sun, Peng, et al. "On the diversity and realism of distilled dataset: An efficient dataset distillation paradigm." CVPR 2024.
>
> [e] Yin, Zeyuan, Eric Xing, and Zhiqiang Shen. "Squeeze, recover and relabel: Dataset condensation at imagenet scale from a new perspective." NeurlPS 2023 Spotlight.
>
> [f] Shao, Shitong, et al. "Generalized large-scale data condensation via various backbone and statistical matching." CVPR 2024 Highlight.

---

> ### Comment · Reviewer_zbwg · 2024-08-08
>
> Thank you for addressing some of my comments in the rebuttal. After reviewing the appendices again, I still do not see how the theoretical analyses can be generalized to two Mixture of Generalized Gaussian distributions (MGG). This requires further discussion.
>
> However, given that you have extended the work to high-resolution images, various tasks, and complex datasets, I am inclined to increase the contribution rating to 4. If you provide more discussion on the theoretical analysis and address all comments in the final draft, I will consider raising my score to a strong acceptance.

---

> > ### Author Response · Authors · 2024-08-10
> >
> > Thank you for your response. We apologize for the oversight regarding MGG. Upon re-evaluating the proofs in our paper, we have determined that our argument does not rely on any specific properties unique to MG compared to MGG. Consequently, transitioning from MG to MGG will mainly involve modifying certain constants and substituting $\Sigma^2$ with $\frac{\alpha^2 \Gamma(3 / \beta)}{\Gamma(1 / \beta)}$ (assuming the probability density function is $\frac{\beta}{2 \alpha \Gamma(1 / \beta)} e^{-(|x-\mu_i| / \alpha)^\beta}$).
> >
> > Below, we present the revised proof of Theorem 1, now based on MGG. For Theorem 2, the only modification is in the setup, which now aligns with the setup used in the proof of Theorem 1 as outlined below. The core reasoning remains consistent with the original proof based on MG, indicating that the proof process detailed in Appendix A and B is generalizable. As for the other theorems in this paper, no updates are required since their proofs do not depend on assumptions regarding the data distribution.
> >
> > Should you have any further questions or require additional clarification, please feel free to reach out.

---

> ### Author Response · Authors · 2024-08-10
>
> **Proof of MGG version of Theorem 1：**
> ## **Setup**
>
> **Notation:** $\textup{N}(\mu,\alpha,\beta)$ denotes the generalized Gaussian distribution with pdf $\frac{\beta}{2 \alpha \Gamma(1 / \beta)} e^{-(|x-\mu| / \alpha)^\beta}$, $\textup{B}$ for Bernoulli distribution.
> We focus on the 1-dim situation. Assume that $\mu_1 < \mu_2$. Define the original data distribution ($\mathcal{N}_0 = \textup{N}(\mu_1, \alpha_0,\beta_0)$ and $\mathcal{N}_1 = \textup{N}(\mu_2, \alpha_0,\beta_0)$):
>
> $$ G := \{(x, y) \mid y \sim 2 \cdot \textup{B}(1,\frac{1}{2}) - 1, x \sim \frac{1-y}{2} \cdot \mathcal{N}_0 + \frac{1+y}{2} \cdot \mathcal{N}_1 \} $$
>
> and the modified one ($\mathcal{N}_0' = \textup{N}(\mu_1, \alpha,\beta)$ and $\mathcal{N}_1' = \textup{N}(\mu_2, \alpha,\beta)$):
>
> $$ G' := \{(x, y) \mid y \sim 2 \cdot \textup{B}(1,\frac{1}{2}) - 1, x \sim \frac{1-y}{2} \cdot \mathcal{N}_0' + \frac{1+y}{2} \cdot \mathcal{N}_1'  \} $$
>
> Our task is predicting $y$ given $x$. Note that $y \in \{\pm1\}$, which is a bit different from the definition in Section 3.2 . In the 1-dim situation, we just need one parameter for this classification task, so define $f_{\theta}(x) := \textup{sign}(x + \theta)$ to fit the distribution.
>
> We could compute the generalization loss on the original distribution:
>
> $$ \mathcal{L}(f _{\theta}) = \left( \int _{-\theta}^{+\infty} \, dF _- + \int^{-\theta} _{-\infty} \, dF _{+} \right) / 2 = \left( 1 - \int _{-\frac{\theta + \mu _2}{\alpha_0}}^{-\frac{\theta + \mu_1}{\alpha_0}} \, dF \right) / 2 $$
>
> Obviously $\theta^\star = -\frac{\mu_1 + \mu_2}{2}$, we have:
>
> $$ \mathcal{L}(f_{\theta}) - \mathcal{L}(f_{\theta^\star}) = \left( \int_{-\frac{\mu_2 - \mu_1}{2 \alpha_0}}^{\frac{\mu_2 - \mu_1}{2 \alpha_0}} \, dF - \int_{-\frac{\theta + \mu_2}{\alpha_0}}^{-\frac{\theta + \mu_1}{\alpha_0}} \, dF \right) / 2 \leq C_1 \cdot (\theta - \theta^\star)^2 \quad (\text{or } C_1' \, |\theta - \theta^\star|) $$
>
> where $C_1$, $C_1'$ are constants, $F_0$, $F_1$, and $F$ denote the CDF of $\mathcal{N} _0$, $\mathcal{N} _1$, and $\textup{N}(0,1,\beta_0)$ respectively.
>
> The inequality above is due to the fact that the function $h(x) = \left( \int^{1}_ {-1} \, dF - \int _{x-1}^{x+1} \, dF \right) / x^2$ has limits at 0 and so is bounded.
>
> ## **Algorithm**
>
> For a dataset $\{(x_i, y_i)\}_{i=1}^{n}$,
>
> set the loss function $L(\theta) = \frac{1}{n} \sum_{i=1}^{n} \ell \left[ y_i (x_i + \theta) \right]$, $\ell(v) = \tfrac{1}{2} (1-v)^2$.
>
> We apply the stochastic gradient descent algorithm and assume the online setting ($n=1$): at step $t$, draw one sample $(x_t, y_t)$ from $G'$ then use the gradient $\nabla L(\theta_t)$ to update $\theta$ ($\eta \in (0,1), t \in \mathbb{N}$):
>
> $$ \theta_{t+1} = \theta_t - \eta \nabla L(\theta_t) $$
>
> $$ \nabla L(\theta_t) = \theta + (x_t - y_t) $$
>
> It can be observed that the randomness of $x$ leads to noise on the gradient.
>
> ## **Bounds with Variance**
>
> We prove the proposition that lower variance of GG can make convergence faster, i.e., $\mathbb{E}\left[ \mathcal{L}(f_{\theta_t}) - \mathcal{L}(f_{\theta^\star}) \right]$ is bounded by an increasing function of the variance ($t$ fixed).
>
> ## **Proof**
>
> From above, we get:
>
> $$ \theta_t = (1 - \eta)^{t} \theta_0 - \eta \left[ (x_{t-1} - y_{t-1}) + (1 - \eta) (x_{t-2} - y_{t-2}) + \dots + (1 - \eta)^{t-1} (x_0 - y_0) \right] $$
>
> and so:
>
> $$ \mathbb{E}\left[ \mathcal{L}(f_{\theta_t}) - \mathcal{L}(f_{\theta^\star}) \right] \leq C_1 \mathbb{E}\left[ (\theta_t - \theta^\star)^2 \right] $$
>
> $$ = C_1 \mathbb{E} \left\( \left[ (1 - \eta)^{t} (\theta_0 - \theta^\star) - \eta \sum_{j=1}^{t} (1 - \eta)^{j-1} (x_{t-j} - y_{t-j} + \theta^\star) \right]^2 \right\) $$
>
> $$ = C_1 \mathbb{E} \left[ (1 - \eta)^{2t} (\theta_0 - \theta^\star)^2 + \eta^2 \sum_{j=1}^{t} (1 - \eta)^{2(j-1)} (x_{t-j} - y_{t-j} + \theta^\star)^2 \right] $$
>
> $$ = C_1 \left\( (1 - \eta)^{2t} (\theta_0 - \theta^\star)^2 + \frac{\eta}{(2 - \eta)} (1-(1 - \eta)^{2t})\left[\frac{\alpha^2 \Gamma(3 / \beta)}{\Gamma(1 / \beta)} + \left( 1 - \frac{\mu_2 - \mu_1}{2} \right)^2\right] \right\) $$
> The last two equalities are due to the fact that for $(x, y) \sim G'$:
>
> $$ \mathbb{E}\left[ x - y + \theta^\star \right] = 0 $$
>
> $$ \mathbb{E}\left[ (x - y + \theta^\star)^2 \right] = \frac{\alpha^2 \Gamma(3 / \beta)}{\Gamma(1 / \beta)} + \left( 1 - \frac{\mu_2 - \mu_1}{2} \right)^2 $$

---

> > ### Comment · Reviewer_zbwg · 2024-08-11
> >
> > Thank you for the detailed proof. I would like to increase my rating to 7.

---

> > > ### Author Response · Authors · 2024-08-12
> > >
> > > We appreciate your insightful feedback and thorough review of our paper. Your comments have been pivotal in enhancing our work and refining the proposed framework. We are committed to addressing any remaining concerns and welcome further discussions or clarifications.

---

### Official Review · Reviewer_fJin · 2024-07-13

**Soundness:** 3
**Presentation:** 3
**Contribution:** 4
**Rating:** 6
**Confidence:** 3

**Summary:**

The authors propose a theoretically motivated dynamic distilled datasets. With use of these transportable representation the authors show that they can outperform the performance of the original dataset and in some cases even the complete dataset.

**Strengths:**

- The methods have a sound theoratical basis, the proofs presented although for a narrow setup our quite interetsing.

- The setup for experiments is well defined.

- The results show extensive performance improvement.

**Weaknesses:**

- The writing style can improve and make the paper approachable. The notations are often terse and lack details about symbols until later. Especially for proofs in appendix.

- It might be more prudent to add experiments of larger scale.

**Questions:**

See weakness

**Limitations:**

The authors did not present a detailed study of where the approach can fail and limitation of their analysis.

---

> ### Author Rebuttal · Authors · 2024-08-07
>
> > [W1] The writing style can improve and make the paper approachable. The notations are often terse and lack details about symbols until later. Especially for proofs in appendix.
> >
>
> [R1] Thank you for your feedback. We will revise the paper to improve its clarity and make it more accessible. Specifically, we will provide more detailed explanations of symbols and notations earlier in the text, particularly in the proofs section of the appendix.
>
> > [W2] It might be more prudent to add experiments of larger scale.
> >
>
> [R2] Thank you for your feedback. The ImageNet-1K dataset utilized in this study is already a substantial and challenging benchmark in the field of dataset distillation [a,b,c,d,e,f,g]. We have conducted additional large-scale experiments, as detailed in attached PDF and our response to Reviewer zbwg.
>
> Our additional experiments detailed in the attached PDF include the following: (1) Utilization of the higher resolution dataset CelebA-HQ (1024 × 1024). (2) Implementation of a continual learning task. (3) Comparisons with more state-of-the-art (SOTA) baselines. (4) Reporting of computation time and peak GPU memory usage. (5) Segmentation task performance on the VOC 2012 dataset. (6) Evaluation on the larger ImageNet-21K dataset.
>
> [a] Sun, Peng, et al. "On the diversity and realism of distilled dataset: An efficient dataset distillation paradigm." CVPR 2024.
>
> [b] Lee, Yongmin, and Hye Won Chung. "SelMatch: Effectively scaling up dataset distillation via selection-based initialization and partial updates by trajectory matching." ICML 2024.
>
> [c] Lee, Dong Bok, et al. "Self-supervised dataset distillation for transfer learning." ICLR 2024.
>
> [d] Chen, Xuxi, et al. "Data distillation can be like vodka: Distilling more times for better quality." ICLR 2024
>
> [e] Yin, Zeyuan, Eric Xing, and Zhiqiang Shen. "Squeeze, recover and relabel: Dataset condensation at imagenet scale from a new perspective." NeurlPS 2023 Spotlight.
>
> [f] Shao, Shitong, et al. "Generalized large-scale data condensation via various backbone and statistical matching." CVPR 2024 Highlight.
>
> [g] Shao, Shitong, et al. "Elucidating the Design Space of Dataset Condensation.” Preprint 2024.

---

> > ### Comment · Reviewer_fJin · 2024-08-09
> > **Thanks**
> >
> > Thank you reviewers for your comments.
> >
> > I will keep my original score.

---

> > > ### Author Response · Authors · 2024-08-10
> > >
> > > Thank you for your prompt response and for taking the time to review our submission. We greatly appreciate your feedback and respect your decision to maintain your original score. Should you have any further comments or require additional clarification, we remain available to address any concerns.

---

### Author Rebuttal · Authors · 2024-08-07

Dear Reviewers,

Thank you again for your constructive comments, which have been very helpful in improving our paper.

During the rebuttal period, we have addressed your concerns in detail in our responses. We have also provided additional experimental results in the attached PDF.

Thank you very much for your precious time and attention.

Best wishes,

Authors of Paper 19675

---

### Decision · Program_Chairs · 2024-09-25

**Decision:**

Accept (poster)

**Comment:**

Some reviewers observe that the theoretical work is on a simplified case. However, as the authors argue effectively in rebuttal, it is valid to perform theoretical analysis on a simplified case, and then use empirical evaluation to determine if the theory does indeed transfer to practice. The rebuttal includes a nice improvement to the theory (generalizing to MMG), but of course that case still remains simple compared to practice.  But, to emphasize: this is not a weakness of the paper per se, the theory remains interesting in its own right, and as a motivation for the empirical method.

The reviewers all agree that the empirical work does indeed show improvements, and the rebuttal's additional experiments show results on additional datasets as requested by the reviewers, including a scale-up to ImageNet-21K.

Reviewer K expresses concern that "the acceleration comes from additional supervision signals, which transforms a self-supervised learning problem to an almost fully supervised one, since a well-trained teacher is involved".  If this were the case, it would reduce the significance of the paper, but the rebuttal argues well that the supervision signal on its own is insufficient, and indeed the observation that "Small-scale CIFAR models improve ImageNet" also rebuts the concern.